# N-terminal proteoforms may engage in different protein complexes

Annelies Bogaert[1,2], Daria Fijalkowska[1,2], An Staes[1,2], Tessa Van de Steene[1,2], Marnik Vuylsteke[3], Charlotte Stadler[4], Sven Eyckerman[1,2], Kerstin Spirohn[5,6,7], Tong Hao[5,6,7], Michael A Calderwood[5,6,7], Kris Gevaert[1,2]

Alternative translation initiation and alternative splicing may give rise to N-terminal proteoforms, proteins that differ at their N-terminus compared with their canonical counterparts. Such proteoforms can have altered localizations, stabilities, and functions. Although proteoforms generated from splice variants can be engaged in different protein complexes, it remained to be studied to what extent this applies to N-terminal proteoforms. To address this, we mapped the interactomes of several pairs of N-terminal proteoforms and their canonical counterparts. First, we generated a catalogue of N-terminal proteoforms found in the HEK293T cellular cytosol from which 22 pairs were selected for interactome profiling. In addition, we provide evidence for the expression of several N-terminal proteoforms, identified in our catalogue, across different human tissues, as well as tissue-specific expression, highlighting their biological relevance. Protein–protein interaction profiling revealed that the overlap of the interactomes for both proteoforms is generally high, showing their functional relation. We also showed that N-terminal proteoforms can be engaged in new interactions and/or lose several interactions compared with their canonical counterparts, thus further expanding the functional diversity of proteomes.

## Introduction

Eukaryotic protein-coding genes give rise to several protein variants, or proteoforms, through various mechanisms including genetic alterations, alternative promoter usage during transcription, and alternative splicing (1, 2, 3). Crosstalk between these mechanisms greatly expands a proteome's complexity (4). Studies in our laboratory revealed that 10–20% of protein N-termini in several human and mouse cells point to alternative translation initiation and/or alternative splicing (5). Such N-terminal proteoforms stem from the same gene but differ at their N-terminus. In eukaryotes,

the canonical mechanism for translation to start involves a ribosome assembling at the 5′ end of a mature mRNA molecule, which then starts scanning for start codons toward the 3′ end. Alternative start codons can be used for translation by various mechanisms such as leaky scanning or internal ribosome entry sites (6, 7). In addition, alternative splicing may give rise to transcripts that have different 5′ ends (e.g., because of skipping of the first exon) (8, 9). Most of the N-terminal proteoforms are truncated at the N-terminus relative to the canonical form; however, up to 6% that have extended N-terminal regions presumably caused translation from codons in the annotated 5′UTR. N-terminal proteoforms can also carry modified N-termini different from those of the canonical protein (2, 10, 11).

N-terminal proteoforms are often overlooked, and information on their biological function is often based on atomistic studies focusing on one gene (2, 9, 12, 13, 14). They may have different functions as the N-terminus of a protein steers several protein features such as half-life and protein localization (15, 16). Of note, mounting evidence indicates that alternative translation initiation is regulated in response to a variety of stress stimuli and/or in a tissue and a cell developmental–specific manner (3, 17, 18). Van Damme et al (2014) also showed that alternative translation initiation sites (TISs) are generally conserved among eukaryotes, hinting at their possible biological impact (5). In addition, several N-terminal proteoforms have already been linked to human diseases, illustrating their potential for therapeutic intervention, and diagnosing and prognosing disease (2, 19).

Other studies showed that N-terminal proteoforms may have altered functionalities (14, 17, 18, 20, 21, 22, 23, 24, 25). Different studies illustrated that N-terminal proteoforms can interact with proteins other than the interaction partners of their canonical protein. For example, the FGF-2 exists in multiple proteoforms: a low molecular weight N-terminal proteoform (18 kD) generated upon alternative usage of a start codon, and at least two higher molecular weight proteoforms (21 and 23 kD) generated upon translation starting from CUG codons located in the 5′UTR of the corresponding transcript. The 18-kD and 23-kD proteoforms have

[1]VIB Center for Medical Biotechnology, VIB, Ghent, Belgium   [2]Department of Biomolecular Medicine, Ghent University, Ghent, Belgium   [3]Gnomixx, Melle, Belgium   [4]Department of Protein Science, KTH Royal Institute of Technology and Science for Life Laboratories, Stockholm, Sweden   [5]Center for Cancer Systems Biology (CCSB), Dana-Farber Cancer Institute, Boston, MA, USA   [6]Department of Genetics, Blavatnik Institute, Harvard Medical School, Boston, MA, USA   [7]Department of Cancer Biology, Dana-Farber Cancer Institute, Boston, MA, USA

Correspondence: kris.gevaert@vib-ugent.be

different localizations and different functionalities. Moreover, the 23-kD FGF-2 proteoform co-immunoprecipitated with the survival motor neuron protein, whereas the 18-kD proteoform did not. The authors hence concluded that the survival motor neuron specifically interacts with the 23-kD FGF-2 proteoform by binding to its N-terminal extension (14).

Protein–protein interactions (PPIs), either stable or transient, are important for cellular functions and regulate cellular signaling (26). Mapping of PPI networks is thus essential to understand cellular processes and signaling pathways, and for defining the origin of several human diseases (27). Several efforts were made to create huge databases that contain experimentally determined PPIs of different organisms, such as BioGRID (28) and STRING (29, 30). As mentioned by Ghadie et al, these databases typically assume that one gene encodes for one protein and ignore the effects of protein modifications, alternative splicing, alternative translation initiation, and other mechanisms leading to proteoforms (31). Recently, a global study revealed a huge impact of protein isoforms originating from alternative splicing on the composition of protein complexes (8) and this often in a tissue-specific way as most proteoforms are expressed in specific tissues and play a role in network organization, function, and cross-tissue dynamics (8, 31, 32). Hence, proteoforms may not be overlooked when studying PPIs.

Based on different reports focusing on single pairs of proteoforms, we hypothesized that different N-terminal proteoforms can be engaged in different protein complexes. As indicated in a recent review (33), no systems-wide information about the interplay between specific proteoforms and protein complex formation is yet available, and as proteins mainly function as part of protein complexes, it is interesting to explore to what extent proteoforms indeed affect interactomes (33). Our prime objective was to assess our hypothesis at a larger scale using a contemporary approach for characterizing protein complexes, and by unraveling the PPIs of N-terminal proteoforms, we aimed to learn more about their functions and how they contribute to the global functional complexity of the proteome. Here, we first applied N-terminal COFRADIC (34) on the cytosol of HEK293T cells to construct a comprehensive catalogue of N-terminal cytosolic proteoforms. N-terminal COFRADIC, in essence, relies on two consecutive, identical chromatographic separations of peptides, interrupted by a chemical reaction with 2,4,6-trinitrobenzenesulfonic acid (TNBS) causing a hydrophobic shift of internal peptides, which is exploited to capture N-terminal peptides during the second chromatographic separation (34). We then applied stringent filtering to select proteoform pairs for interactome analysis by Virotrap (see Fig 1). In short, in Virotrap, a bait protein is fused to the C-terminus of the HIV-1 GAG protein, leading to the recruitment of the GAG–bait fusion protein at the plasma membrane where GAG multimerization occurs, followed by subsequent budding of virus-like particles (VLPs) from the cells. As the bait is coupled to GAG, this allows for co-purification of bait-associated protein partners by trapping them into VLPs. Purification of the VLPs themselves relies on co-expressing FLAG-tagged and untagged VSV-G, presented as trimers on the surface of VLPs, allowing for efficient antibody-based purification of the VLPs. Of note, Virotrap was shown to be a sensitive PPI method, as VLPs encapsulate and preserve the protein complexes, allowing the detection of weak and transient PPIs (35).

In this study, we identified 3,306 protein N-termini in the cytosol of HEK293T cells of which 1,044 originate from N-terminal proteoforms, highlighting the prevalence of both alternative translation initiation and alternative splicing. We provide evidence for the existence of several of these N-terminal proteoforms in other cells and tissues, supporting their biological relevance. Virotrap-based interactome analysis of 20 carefully selected pairs of N-terminal proteoforms and their canonical protein revealed that N-terminal proteoforms not only share most of their interactions with the canonical protein, yet also have their own set of unique interaction partners, whereas other interactions can be lost.

# Results

## Construction of an N-terminal proteoform catalogue of the HEK293T cellular cytosol

To study the interactome of N-terminal proteoforms, we first constructed a catalogue of N-terminal proteoforms of cytosolic proteins in HEK293T cells as Virotrap currently only functions in these cells and favors cytosolic proteins as baits. Such a decrease in proteome sample complexity also increases the possibility of identifying N-terminal proteoforms (36). Both Western blot (WB) and gene ontology cellular component (GOCC) data analysis indicated that we strongly enriched for cytosolic proteins (see Fig S1A and B). Given that N-terminal proteoforms have different N-termini, we enriched for N-terminal peptides of the cytosolic proteins by N-terminal COFRADIC (omitting the SCX pre-enrichment step) (34). In parallel, three different proteases, trypsin, chymotrypsin, and/or endoproteinase GluC, were used as this further increases the depth of analysis (37).

The LC-MS/MS data were searched using the UniProt database (restricted to human proteins) and a custom-built database, which included all human UniProt proteins and UniProt isoforms, supplemented with protein sequences built from two HEK293T Ribo-seq datasets, and contains 103,020 non-redundant protein sequences (36). As the custom database is significantly larger than the UniProt database (which only holds 20,356 proteins), and this negatively affects the false discovery rate (FDR) and the number of identified proteins (38, 39), we opted to combine the data of the two searches, thus boosting the total number of identifications. We evaluated how efficient N-terminal COFRADIC enriched for N-terminal peptides and found enrichment up to 60%, which is similar as previously reported (34) (see Fig S1C). Similar as reported in Reference (36), the enrichment efficiency for chymotrypsin-digested samples is lower.

Further bioinformatics data curation and analysis, facilitating the selection of N-terminal proteoform pairs for interactome analysis, is summarized in Fig 2A. First, we applied stringent filtering on the identified peptides to retain N-terminal peptides originating from translation, as previously described (36) (see Fig 2, step 3, and its details in Fig 2B). We started by selecting N-termini pointing to database-annotated translation start sites and continued with inspecting N-terminal peptides starting upstream or downstream of annotated start sites to identify N-terminal proteoforms.

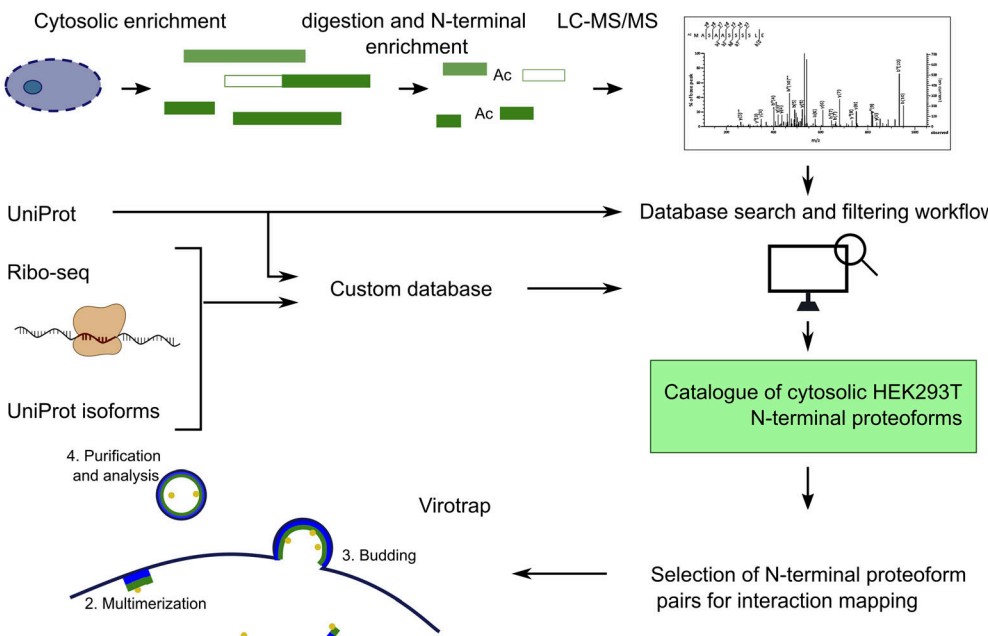

**Figure 1. Overview of the experimental approach.**
HEK293T cells are grown followed by digitonin lysis to enrich for cytosolic proteins. Proteins are subsequently digested (in parallel using different proteases), and protein N-terminal peptides are enriched by COFRADIC, analyzed by LC-MS/MS, and searched with two different databases: the UniProt database and a custom-built database also including UniProt isoforms and ribosome profiling (Ribo-seq) data. The data are then thoroughly filtered to generate a catalogue of N-terminal proteoforms, and from this, pairs of canonical protein and N-terminal proteoforms are selected for interactome analysis by Virotrap.

Evaluation of candidate alternative N-termini favors peptides with co-translational N-terminal acetylation, also considering its interplay with initiator methionine removal by methionine aminopeptidases and supported by extra translational evidence provided by Ribo-seq (included in the custom search).

By this filtering strategy, we only retained confident N-terminal peptides pointing to database-annotated protein starts or N-terminal proteoforms, and assigned a confidence level to these peptides. As a last step, N-terminal peptides identified by different proteases were merged into a final dataset (see Fig 2) of 3,306 unique N-terminal peptides (see Table 1 and Table S1). We show that the combination of data from three different proteases increases the proteome coverage (see Fig S1D).

As can be seen in Table 1 and Table S1, most of the identified peptides are known N-terminal peptides listed in UniProt (2,262 or 68.4%), whereas the remaining 1,044 (31.6%) N-terminal peptides point to potential (N-terminal) proteoforms. From these potential N-terminal proteoforms, 874 N-terminal peptides were identified that stem from translation starting at an internal site of a UniProt canonical protein. We also identified 90 N-terminal peptides only matching a UniProt isoform and 80 N-terminal peptides only matching an Ensembl entry (a proteoform translated from Ensembl transcript and derived from Ribo-seq evidence). N-terminal peptides matching non-canonical accessions often originate from N-terminal proteoforms as these Ensembl and UniProt isoform accessions include splice variants (with differences at the N-terminus) or the products of alternative translation initiation.

### Selection of N-terminal proteoforms for interaction mapping

To trim down the catalogue of 1,044 N-terminal proteoforms to a more manageable set of pairs of N-terminal proteoforms and corresponding canonical proteins to test by Virotrap, we gathered extra information on the identified proteins to select confident and potentially interesting pairs.

As Virotrap is currently restricted to cytosolic proteins, it is important to verify the cytosolic localization of both the canonical protein and its N-terminal proteoform. Therefore, we generated a map of cytosolic proteins in HEK293T cells. Cytosolic extracts were prepared with 0.02% digitonin of HEK293T cells in triplicate, and after trypsin digestion and peptide pre-fractionation, LC-MS/MS analysis led to the identification of 3,045 proteins (Table S2). Besides the gene ontology (GO) terms added in the Perseus analysis, all identified proteins were submitted to the Retrieve/ID Mapping tool on the UniProt website to evaluate their associated GOCC terms and around 60% of these proteins contain the GO term "cytosol" (GA:0005829). When considering label-free quantitation (LFQ) intensities of these cytosolic proteins, we see that they account for 86% of the sample. Note that this map will be used for the selection of proteoform pairs.

To gather more gene- and protein-centered information, an R workflow was developed (see Fig 2A) to extract genomic information about the proteoforms (such as genomic coordinates, exons, start codon, frame, gene, biotype, transcript support levels) and find protein sequence features (such as known domains and eukaryotic linear motifs [ELMs]) that can be lost/gained by proteoforms of the same gene. We supplemented this information with disease association from the OMIM database, known interaction partners in BioGRID, whether or not the N-terminal peptide was previously reported (5), and cytosolic localization reported in our own cytosolic map (see above), the Human Protein Atlas, and gene ontology GOSlim annotation. Based on this extra information, including lost protein domains, different predicted linear motifs, the use of non-AUG start codons, the presence of co-translational modifications,

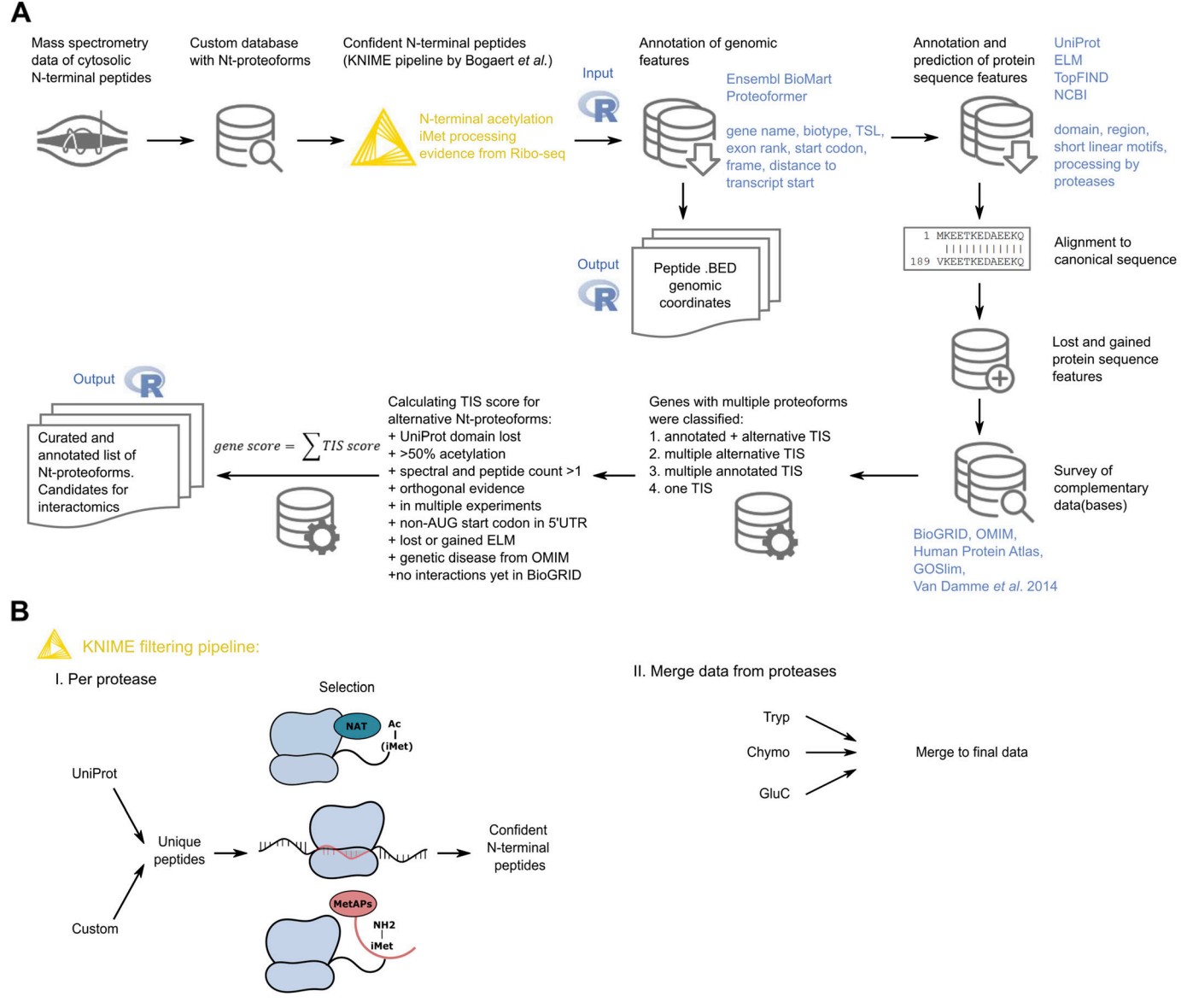

**Figure 2. Selection of candidate N-terminal proteoforms.**
**(A)** N-terminal proteomics data curation and analysis workflow included a KNIME pipeline for stringent filtering of N-terminal peptides (see Bogaert et al (43) and Fig 2B), followed by annotation of genomic and protein sequence features of candidate N-terminal proteoforms using a custom R workflow. Based on the collected information, a translation initiation site score and its derivative gene score were calculated to prioritize genes with alternative N-terminal proteoforms for interactome validation. **(B)** Overview of the N-terminal peptide filtering workflow. Per protease, the data are merged and filtered to obtain unique peptide sequences before selection of confident N-terminal peptides originating from translation. This selection is done based on co-translational acetylation of a protein's N-terminus, extra translational evidence by Ribo-seq, and the presence or potential processing of the initiator methionine by methionine aminopeptidases. Confident N-terminal peptides found for each protease-treated sample were merged to obtain a dataset of N-terminal peptides.

higher spectral and peptide counts, and orthogonal evidence from UniProt, Ensembl, or Ribo-seq, we built a scoring system considering the most relevant parameters listed for each peptide, called TIS scoring. TIS scores of the different N-terminal peptides identified from the same gene were then combined into a gene score (see Fig 2A). Most importantly, when there is evidence that the detected peptide is not an N-terminal peptide resulting from translation but rather from processing (potential dipeptidase, signal/transit/propeptide processing, or potential cleavage site of a protease), TIS scores drop to zero. The resulting list of N-terminal peptides including all extra information and TIS/gene scores is shown in Table S3. In total, 372 genes (corresponding to 868 N-terminal peptides) have a gene score > 0 (see Table S3, second tab).

To select pairs of canonical proteins and N-terminal proteoforms for interactome profiling, we developed a prioritization strategy based on protein localization, expression, and other functional and structural parameters (see below). To gather more information

**Table 1. Overview of all identified N-terminal peptides after applying stringent selection.**

|  | Combined results |
|---|---|
| UniProt proteins | 3,136 |
| Database-annotated | 2,262 |
| Alternative TIS | 874 |
| High-confident | 444 |
| Low-confident | 430 |
| UniProt isoforms | 90 |
| Database-annotated | 77 |
| Alternative TIS | 13 |
| High-confident | 7 |
| Low-confident | 6 |
| Ribo-seq–identified (ENST) | 80 |
| TIS in database-annotated CDS | 39 |
| TIS in 5′UTR | 22 |
| TIS in non-translated regions (NTR) | 19 |
| Total | **3,306** |
| Database-annotated TIS | 2,262 |
| Alternative TIS | 1,044 |

N-terminal peptides are classified based on the type of protein sequence they are linked to, being a regular UniProt protein, a UniProt isoform, or a Ribo-seq–derived protein sequence. A distinction is made between database-annotated N-terminal peptides (starting at position 1 or 2) and peptides with a start position beyond two, thus pointing to N-terminal proteoforms (alternative translation initiation start site). Based on translational evidence, a confidence level is assigned (either low or high confident) to the identified N-terminal proteoforms.

about their expression, we examined whether the identified N-terminal proteoforms are also expressed in other cell lines or tissues as this expands their biological significance. This analysis was done by exploiting WB data present in the Human Protein Atlas for these 372 genes. To detect N-terminal proteoforms on WB, clearly the difference in molecular weights between the canonical protein and N-terminal proteoform needs to be sufficiently large to be detectable, and the epitope targeted by the antibody needs to be preserved in the proteoform, which reduced the list to 138 genes. For each of these, information on all tested antibodies (also unpublished ones) was extracted and antibodies with a WB score of 2, which indicates the detection of a protein band of the predicted size (±20%) with additional bands present, were selected. In total, data on 621 antibodies detecting the protein products of 136 genes were retrieved. Note that for two genes, no antibody information was present. For 143 of the 621 antibodies, extra bands were reported and the blots with these antibodies (detecting the protein products of 95 genes) were evaluated in more detail (online at www.proteinatlas.org, under antibodies and validation). For the protein products of 34 genes, an extra band corresponding to the size of the N-terminal proteoform(s) was detected, indicating a plausible expression of these N-terminal proteoforms in other cell lines and/or tissues (see Table S4). For example, a band corresponding to the N-terminal proteoform of FNTA (40.4 kD instead of

44.4 kD, antibody CAB010149) was found in RT4 and U-251 MG cell lines and in liver and tonsil tissues.

To further evaluate the expression of N-terminal proteoforms in healthy human tissues, we re-analyzed public proteomics data of the draft map of the human proteome developed by the Pandey group (40). The use of ionbot (41 *Preprint*) and a custom-built protein sequence database (composed of UniProt- and Ribo-seq–derived proteoforms) led to 9,151,086 peptide-to-spectrum matches (PSMs). Further filtering and aggregation of the data was performed in R, leading to 8,501,009 filtered PSMs and 2,789,079 unique peptides belonging to 26,159 proteoforms. Of the 3,306 proteoforms identified by N-terminal COFRADIC, 897 were found expressed in human tissues and supported by an N-terminal peptide identified at a matching start position. Among these 897 proteins, 24 are non-canonical N-terminal proteoforms, thus proteoforms with peptides that do not match a canonical UniProt protein (Fig 3A, Table S5). Differential expression analysis across tissues using normalized spectral abundance factors (NSAF) indicated that 582 of 897 proteoforms matching N-terminal COFRADIC data had a significant tissue-dependent expression profile, including three non-canonical proteoforms of two genes, namely, TPM3 and EPB41L3 (Fig 3B and C). These data confirm that N-terminal proteoforms are not only expressed in histologically healthy human tissues, but also display tissue specificity (Fig 3A) or different tissue expression profiles when comparing N-terminal proteoforms of the same gene (Fig 3B and C).

For the final selection of candidates for interactome profiling by Virotrap, we applied additional criteria for prioritization. First, we prioritized proteoforms with a loss of a known domain or eELM. Then, we prioritized proteoforms with a considerable length of truncation or extension (>20 amino acids or >50 amino acids for proteins over 700 amino acids). Next, we prioritized proteoforms suited for Virotrap, being cytosolic proteoforms (by checking our cytosolic map and the subcellular localizations listed in UniProt and the Human Protein Atlas), and non-structural proteins and enzymes over structural proteins. Finally, we prioritized known disease-associated proteins and proteoforms over novel proteins (e.g., proteins from out-of-frame translation) and considered the results of the Human Protein Atlas–WB analysis.

As a final result, we report 85 proteoform pairs meeting several selection criteria, of which the 22 highest ranking (best scoring over all the criteria) were selected for analysis by Virotrap (see Tables 2 and S3 [third worksheet]). Note that not for all proteins, the canonical N-terminus was identified. This was the case for CAST, CSDE1, SPAST, and UBXN6. In these cases, verification of the cytosolic localization is important as it is known that N-terminal proteoforms and their canonical proteins can have different subcellular localizations (42). Note also that the results of the protein originating from a presumed non-translated region (NTR, ACTB pseudogene 8) have been published before (36) and will thus not be further discussed here. When checking the 175 N-terminal proteoforms corresponding to 85 genes considered for Virotrap experiments in the re-analyzed public proteomics data of the draft map of the human proteome, we found 100 N-terminal proteoforms expressed in human tissues (see Table S5 column "virotrap_intersect").

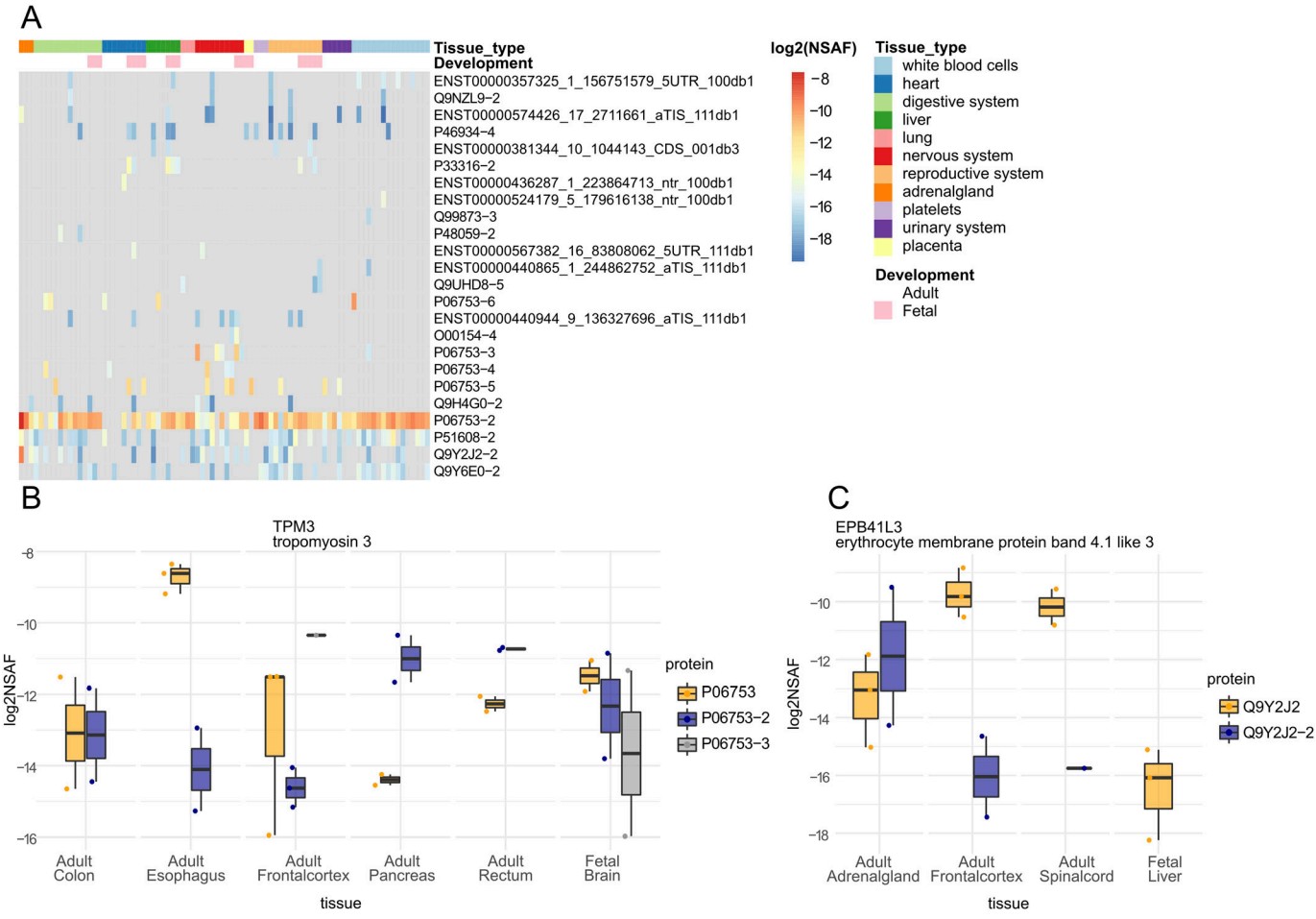

**Figure 3. Tissue expression of N-terminal proteoforms.**
**(A)** Heatmap presenting log$_2$ NSAF expression values of 24 non-canonical proteoforms across healthy human tissues from the Pandey dataset (40). **(B, C)** N-terminal proteoforms from the same gene (TPM3 in panel (B) and EPB41L3 in panel (C)) show distinct profiles of tissue expression. All presented proteoforms were found significant in our differential expression analysis (adjusted $P$-value ≤ 0.05).

## Interaction profiling of N-terminal proteoforms and their canonical counterparts

Mapping of the protein complex a protein is engaged in is a frequently used approach to gain insights into the processes and pathways that protein is involved in (the "guilt-by-association" approach) (27). However, most interactome studies are restricted to studying the canonical protein, but as (N-terminal) proteoforms can have altered functions, including these proteoforms in interactomics studies would help us to better understand their function and how this function is possibly different from that of the canonical protein. For interactome analysis, Virotrap is used. In short, a bait protein is fused to the C-terminus of GAG, leading to the recruitment and multimerization of the GAG–bait fusion protein at the plasma membrane followed by subsequent budding of VLPs from the cells. As the bait is coupled to GAG, this allows for co-purification of bait-associated protein partners by trapping them into VLPs. Purification of the VLPs relies on co-expressing FLAG-tagged and untagged VSV-G, presented as trimers on the surface of VLPs, allowing for efficient antibody-based purification of the VLPs.

To study the interactomes of pairs of canonical proteins and N-terminal proteoforms, we designed the following strategy (Fig 4). First, both canonical proteins (full length, further referred to as FL) and N-terminal proteoforms (referred to as PR) were cloned into pMET7–GAG–bait plasmids where they are N-terminally fused to the GAG protein. This led to 45 pMET7–GAG–bait constructs of 21 genes with for each gene one FL bait and one or two PR baits (Fig S2A). In an initial Virotrap screen, bait expression was first tested on WB (Fig 4A) to evaluate whether baits were well expressed, FL and PR baits had comparable expression levels, and they were efficiently recruited into VLPs. Note that similar expression levels of the control bait (eDHFR), FL, and PR are desired for the statistical analysis of the LC-MS/MS data (Fig 4A). Baits that were not recruited into VLPs were left out for further analysis, which was the case for CAST and RARS. For CAST, both FL and PR were not pulled into VLPs, whereas for RARS, only the FL was pulled into the VLPs (Fig 4B, uncropped WBs are shown in Fig S2B), not allowing for comparative analysis. All other baits (e.g., PAIP1, Fig 4B) were clearly detected in the cell lysates and in the VLPs.

**Table 2. List of the proteins that were selected for interactome analysis.**

| Protein | Gene | Proteoform | Lost or gained domains and motifs |
|---|---|---|---|
| Q12904 Aminoacyl-tRNA synthetase complex–interacting multifunctional protein 1 Length: 312 AA | AIMP1 | Known UniProt isoform Q12904-2; N-terminal extension of 24 AA | Gain of three linear motifs according to ELM: binding motif for UBA3 adenylation (AA 17–24), SH3 ligand (AA 4–10), and SPAK/OSR1-docking motif (AA 14–18) |
| Q9HB71 Calcyclin-binding protein Length: 228 AA | CACYBP | Known UniProt isoform Q9HB71-3; N-terminal truncation of 43 AA. | Partial loss of region for interaction with SIAH1 (AA 2–80) |
| Q14444 Caprin-1 Length: 709 AA | CAPRIN1 | High-confident N-terminal proteoform; N-terminal truncation of 116 AA | Loss of the coiled-coil domain (60–94) |
| P20810 Calpastatin Length: 708 | CAST | The identified N-terminal peptide cannot differentiate which proteoform was identified: known UniProt isoforms P20810-4 (missing AA 9-30 and AA 44–62) or P20810-8 only missing AA 9–30. | — |
| Q9Y281 Cofilin-2 Length: 166 AA | CFL2 | Known UniProt isoform Q9Y281-3; N-terminal truncation of 17 AA | Partial loss of the region for interaction with CSRP3 (AA 2–55) and partial loss of the ADF-H domain (AA 4–153). |
| O75535 Cold shock domain–containing protein E1 Length: 798 AA | CSDE1 | High-confident N-terminal proteoform; N-terminal truncation of 176 AA | Loss of the CDS1 domain (AA 26–87) and loss of most of the CSD2 domain (AA 136–179) |
| P60842 Eukaryotic initiation factor 4A-I Length: 406 | EIF4A1 | High-confident N-terminal proteoform; N-terminal truncation of 211 AA | Partial loss of the helicase ATP-binding domain (63–234), loss of a Q motif (AA 32–60), and loss of a DEAD box motif (AA 182–185) |
| P49354 Protein farnesyltransferase/geranylgeranyltransferase type-1 subunit alpha Length: 379 AA | FNTA | High-confident N-terminal proteoform; N-terminal truncation of 38 AA | Loss of a Pro-rich region (AA 22–31) |
| Q7Z434 Mitochondrial antiviral-signaling protein Length: 540 AA | MAVS | Known UniProt isoform Q7Z434-4; N-terminal truncation of 141 AA | Loss of the CARD domain (AA 10–77), loss of the region for interaction with NLRX1 (AA 10–77), partial loss of the Pro-rich region (AA 103–153), and the region for TRAF2 interaction becomes outer N-terminal |
| Q9H074 Polyadenylate-binding protein–interacting protein 1 Length: 479 AA | PAIP1 | Known UniProt isoform Q9H074-3; N-terminal truncation of 112 AA | Loss of the Gly-rich region (AA 10–36), loss of the Pro-rich region (AA 45–98), and PABPC interaction motif 2 becomes N-terminal (AA 116–143) |
| Q14558 Phosphoribosyl pyrophosphate synthase–associated protein 1 Length: 356 AA | PRPSAP1 | Known UniProt isoform Q14558-2; N-terminal extension of 29 AA | Gain of 11 linear motifs according to ELM |
| Q86TP1 Exopolyphosphatase PRUNE Length: 453 AA | PRUNE | The identified N-terminal peptide cannot differentiate which proteoform was identified: known UniProt isoforms Q86TP1-3 (missing AA 1–182) and Q86TP1-5 (missing AA 1–182 and AA 259–311) | Loss of a DHH motif (AA 106–108) |
| P49023 Paxillin Length: 591 | PXN | Known UniProt isoform P49023-4; N-terminal truncation of 133 AA and also missing AA 278–311 | Loss of a LD motif (AA 3–15) and loss of the Pro-rich region (AA 46–53) |
| P54136 Arginine-tRNA ligase, cytoplasmic Length: 660 AA | RARS | Known UniProt Isoform P54136-2; N-terminal truncation of 72 AA | Loss of a region that could be involved in the assembly of the multisynthetase complex (AA 1–72) |
| Q9UBP0 Spastin Length: 616 AA | SPAST | The identified N-terminal peptide cannot differentiate which proteoform was identified: known UniProt isoforms Q9UBP0-3 (missing AA 1–86) or Q9UBP0-4 (missing AA 1–86 and AA 197–228) | Loss of the region for interaction with ATL1 (AA 1–80), loss of the region needed for nuclear localization (AA 1–50), loss of the region for interaction with SSNA1 and microtubules (AA 50–87), loss of NLS (AA 4–11), loss of nuclear export signal (AA 59–67), partial loss of the region for interaction with RTN1 (AA 1–300), and partial loss of the region for midbody localization (AA 1–194) |

**Table 2.   Continued**

| Protein | Gene | Proteoform | Lost or gained domains and motifs |
|---|---|---|---|
| Q99576 TSC22 domain family protein 3 Length: 134 AA | TSC22D3 | High-confident N-terminal proteoform; N-terminal truncation of 57 AA. | Almost complete loss of the AP1-binding region |
| Q9BSL1 Ubiquitin-associated domain–containing protein 1 Length: 405 AA | UBAC1 | High-confident N-terminal proteoform; N-terminal truncation of 102 AA | Loss of the ubiquitin-like domain (AA 14–98) |
| P61081 NEDD8-conjugating enzyme Ubc12 Length: 183 AA | UBE2M | High-confident N-terminal proteoform; N-terminal extension of 42 AA. | Gain of a caspase cleavage motif (AA 14–18), gain of a glycosaminoglycan attachment site (AA 13–16 and 35–38), gain of a IAP-binding motif (AA 1–4 and 17–21), and gain of a TRAF2-binding site (AA 3–6) |
| Q9BZV1 UBX domain–containing protein 6 Length: 441 AA | UBXN6 | Known UniProt isoform Q9BZV1-2; N-terminal truncation of 53 AA | Loss of the region for interaction with LMAN1 (AA 1–10) and a VCP/p97-interacting motif (VIM) becomes N-terminal, with 2 AA removed (AA 51–63) |
| P09936 Ubiquitin carboxyl-terminal hydrolase isozyme L1 Length: 223 AA | UCHL1 | Two high-confident N-terminal proteoforms: N-terminal truncations of 5 or 11 AA | Proteoform 1: ubiquitin interaction domain (AA 5–11) becomes outer N-terminal Proteoform 2: loss of the ubiquitin interaction domain (AA 5–11) |
| Q8TCF1 AN1-type zinc finger protein 1 Length: 268 | ZFAND1 | Known UniProt isoform Q8TCF1-2; N-terminal truncation of 107 AA | Loss of the AN1-type 1 zinc finger domain (7–58) and loss of most of the AN1-type 2 zinc finger domain (AA 61-110) |
| ENST00000403258 ACTB pseudogene 8 Length: 146 AA | NTR protein | High-confident N-terminal proteoform; N-terminal truncation of 11 AA | ELM predicts several motifs and domains for this protein. |

For each protein, information about the canonical protein is provided (UniProt protein accession and gene name), and about the N-terminal proteoform(s) concerning the length of the truncation or extension. In the last column, differences between the canonical proteins and the N-terminal proteoforms with regard to domains, motifs, or ELMs are listed.

For LC–MS/MS analysis, we divided the baits into manageable sets including control samples and maximally three pairs of FL and PR. Here, baits with similar expression levels were combined and triplicate experiments for all baits were performed. In total, baits were divided into seven sets (see the Materials and Methods section). To obtain specific interaction partners of the FL or PR from the lists of identified proteins, their interactomes were compared with those of the control baits. To evaluate possible functional differences between FL and PR, the identified proteins using the FL baits were directly compared with those from the PR baits (Fig 4B). Our experiments show that besides the interaction partners and baits, many "background proteins" are consistently identified in the VLPs. In fact, for example in the first set, 458 of 1997 (23%)proteins are identified in 80% of Virotrap samples, providing a basis for MaxLFQ normalization.

Triplicate Virotrap experiments were performed for the protein products of 20 genes (42 proteoforms) taking along GAG-eDHFR as a control. In the following, we illustrate the employed data analysis strategy for MAVS, and this strategy is applied to all baits in the same way. MAVS is annotated as a mitochondrial antiviral-signaling protein for which a N-terminal proteoform starting at methionine-142 was identified, which is a known UniProt isoform. This proteoform loses a CARD domain, which is required for interaction with NLRX1, and thereby, the region required for interaction with TRAF2 becomes outer N-terminal (Fig 5A).

MaxQuant searches of the experimental set including MAVS FL and PR led to the identification of 2,134 unique proteins. Further data analysis was handled in Perseus, and after removal of contaminants, reversed proteins, and proteins only identified by site,

1,997 proteins remained. The protein LFQ intensities were $\log_2$-transformed, and replicates were grouped. Proteins identified in less than three samples in at least one group were removed, leading to a final set of 842 proteins. The missing values were imputed using imputeLCMD. As already mentioned, the bait levels are ideally very similar allowing a straightforward comparison of the levels of interaction partners or commonly co-purified proteins between the different bait interactomes (43). Therefore, as a first check, bait intensities were visualized in a profile plot (before imputation) and both MAVS FL and PR had comparable intensities; however, they both are less intense compared with the control (eDHFR) intensities (Fig 5B).

For 13 baits, CACYBP, CAPRIN1, CSDE1, EIF4A1, FNTA, MAVS, PAIP1, PRPSAP1, UBAC1, UBE2M, UBXN6, UCHL1, and ZFAND1, FL and PR(s) showed similar expression levels. For the seven other baits, AIMP1, CFL2, PRUNE, PXN, NTR, SPAST, and TSC22D3, a difference in expression levels of at least twofold was found between the FL and PR, hampering conclusive statistical analysis (Fig S3). Data of all baits are here reported. Subsequently, all baits were pairwise tested against the eDHFR control samples to identify their candidate interaction partners at a FDR ≤ 0.01, which resulted in 43 candidate interaction partners for MAVS FL and 48 for MAVS PR (Table 3). We compared such lists of potential interaction partners with known interaction partners listed in BioGRID (50), STRING (51), and IntAct (52). Among the candidate MAVS interaction partners, we found BAG6, IFIT1, IFNB1, and TRAF2 (only reported for the FL), which are known interaction partners of MAVS. Of the 43 candidate interaction partners of the canonical protein, 39 (90.7%) are also found using the N-terminal proteoform as a bait. Four proteins seem to be

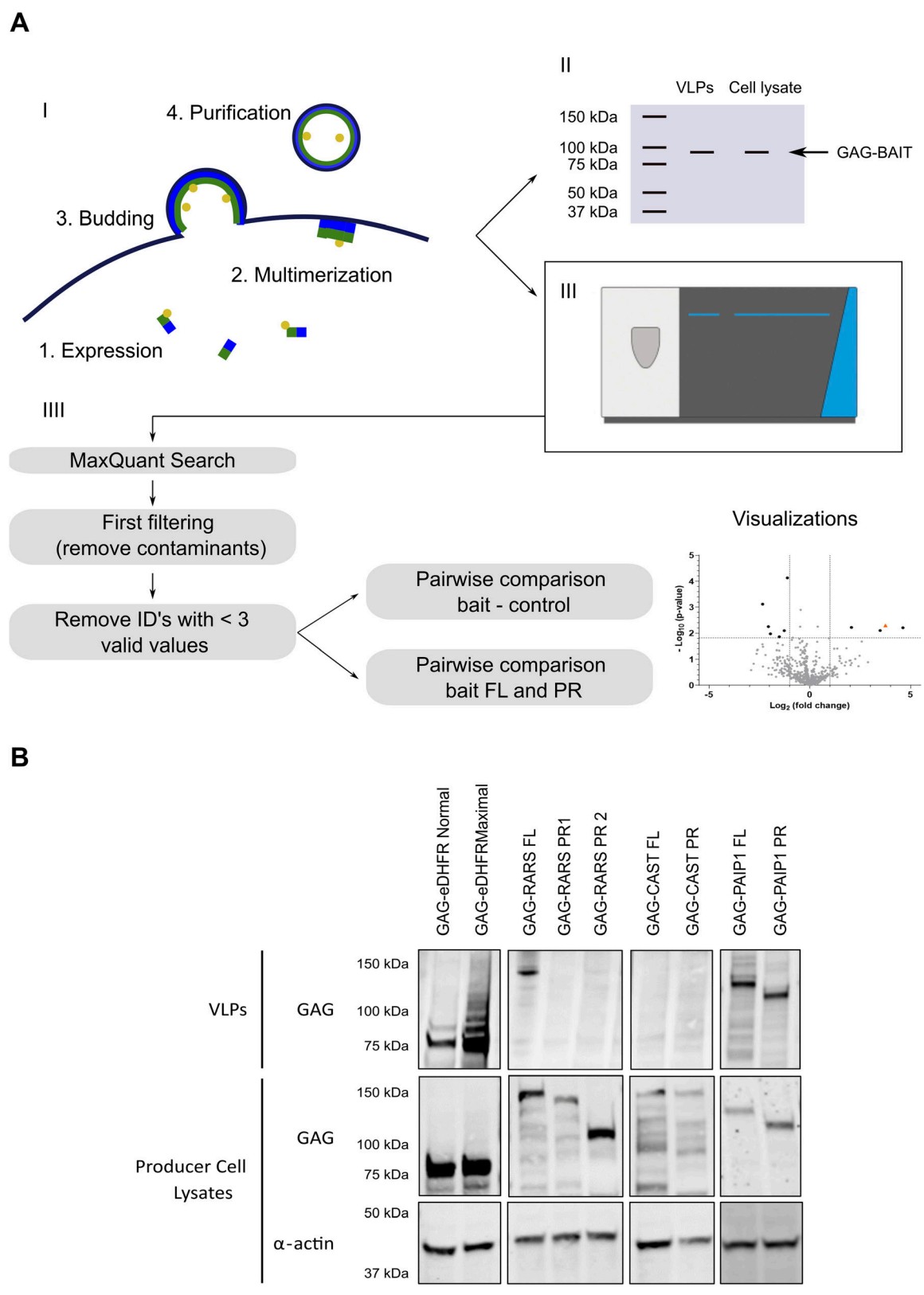

**Figure 4. Overview of the Virotrap workflow.**
**(A)** Protein complex purification by Virotrap and data analysis approach. I. GAG–bait fusion proteins were expressed in HEK293T cells, and after multimerization at the plasma membrane, GAG–bait fusion proteins and their interacting proteins are trapped in VLPs that bud from the cells. These VLPs are subsequently purified from the media and prepared for either WB or LC-MS/MS analysis. II. As an initial screen, all baits are analyzed by WB to validate proper expression and recruitment into VLPs. III.

potential interaction partners for full-length MAVS, ACLY, EEF2, PKM, and TRAF2, whereas nine proteins, ACTL6A, CAD, DPYSL2, PGM1, PKRKACB, PLK1, PRKDC, PYGL, and SEPT7, seem to be specific for the shorter MAVS proteoform, possibly pointing to functional diversities of these two MAVS variants.

For all baits tested in Virotrap, besides AIMP1, we were able to detect known interaction partners among the candidate interaction partners, validating our approach. All identified candidate interaction partners are listed in Table S6. In general, we found high overlaps between the interactomes of paired proteoforms, and on average, 66.7% of the candidate interaction partners of one proteoform were also found for the other proteoform, indicating that most interactors seem to be shared by the different proteoforms. For example, for cases such as UBXN6 (Fig 6, top row, left panel), most of the candidate interaction partners are shared between the proteoforms, yet for each proteoform, unique interactors were found. On the contrary, for some other bait proteoform pairs, the overlap between interactors is quite limited, as shown for PRUNE (Fig 6, top row, middle panel) where only 25% of the candidate interaction partners are shared. On the contrary, for UBAC1 (Fig 6, top row, right panel), whereas the N-terminal proteoform has lost several interactors, it does not seem to engage in other PPIs.

As mentioned above, differences in candidate interaction partners between the canonical protein and N-terminal proteoform may point to functional diversities between protein variants. We aimed to study such differences in more detail by a direct pairwise comparison between the FL and PR interactomes, and withheld proteins with an at least twofold change in levels (also correcting for differences in bait levels; see the Materials and Methods section) and with a pairwise test adjusted P-value (FDR) ≤ 0.05. Furthermore, only proteins already listed as candidate interactors upon comparing with the interactomes of the control samples or already known as interaction partners as listed in BioGRID (50), STRING (51), and IntAct (52) were retained.

The results of all pairwise comparisons between the different proteoforms of all baits are listed in Table S7 and visualized in Fig S4. For all proteins, we report differences between the FL and PR interactomes, and list if a prey was previously reported as a candidate interaction partner. In some comparisons, we report GAG among the significant proteins at a FDR ≤ 0.05. However, except for SPAST FL versus PR1 and SPAST PR1 versus PR2 interactomes, GAG is removed when also filtering on fold change. We hypothesize that GAG might pop up because of differences in expression between FL and PR baits, but by our additional filtering on fold change, we removed GAG, showing the necessity of this additional filtering step to retain reliable differences

and not differences only because of experimental variations. Along with GAG, for some cases, we report known GAG interaction partners among the significant proteins, but mostly not with a sufficiently large fold difference, which illustrates these double filtering steps as a valid way for identifying proteoform-specific interactors.

## Proteoform-specific interaction partners

Some selected findings on proteoform-specific interactors are discussed in the following section.

For MAVS, a pairwise test between the interactomes of FL and PR resulted in 10 significant proteins (Fig 7A): PTK7, PLK1, HNRNPUL2, CEP55, CD2AP, ARRDC1, DAG1, EIF3K, TRAF2, and NUP205. TRAF2 and PLK1 have been reported above as candidate interaction partners for MAVS FL and PR, respectively. This is also obvious from their intensity profiles in the different samples before imputation (Fig 7B), which show that TRAF2 and PLK1 interact with MAVS with a significant difference in intensity in the FL and PR interactomes. For comparison, we also show the profile of two proteins that were reported as significantly different between the FL and PR interactomes, but were removed as they were not listed as candidate interactors. A first example is DAG1, which was not identified in any of the MAVS interactomes, but the difference in the interactomes of the MAVS proteoforms is due to differences in imputed intensity values. In fact, this highlights an imputation-based shortcoming of data analysis; however, imputation is necessary for statistical analysis. A second example is ARRDC1, which is identified in almost all samples with an apparent lower intensity in the MAVS samples, making it thus unlikely that ARRDC1 is an interaction partner of MAVS. To conclude, it seems that MAVS PR does not interact with TRAF2, which could be due to the fact that the domain required for this interaction is exposed at the N-terminal part of MAVS PR (Fig 5A). On the contrary, MAVS PR seems to interact stronger with PLK1, suggesting that MAVS PR can both lose and gain interactors.

For the polyadenylate-binding protein–interacting protein 1 (PAIP1), an N-terminal proteoform starting at position 113, which is a known UniProt isoform, was detected. This PAIP1 N-terminal proteoform specifically interacts with GIGYF2, ZNF598, and EIF4E2, which together form the 4EHP-GYF2 complex. GIGYF2 and ZNF598 were identified from the comparison of the FL and the PR interactomes, whereas ZNF598 and EIF4E2 were listed as candidate interactors of PAIP1 PR (Fig 7C). The engagement of PAIP1 PR with the 4EHP-GYF2 protein complex could point to a different functionality of the PAIP1 N-terminal proteoform versus full-length PAIP1.

---

Baits that are successfully recruited into VLPs (as determined by WB) are suited for further experiments. For LC-MS/MS, three biological replicates of all baits were performed. Baits were divided into manageable sets of one control bait and maximally three pairs of FL and PR. IV. The LC-MS/MS data were searched with MaxQuant's Andromeda search engine against the human proteome (supplemented with the sequences of the proteins expressed for Virotrap such as VSV-G with only the shortest sequence of the baits) and filtered (to remove reversed matches, proteins only identified by site, and contaminants), and the LFQ intensities were transformed. Afterward, samples were grouped and identified proteins were filtered on three valid values in at least one group and subsequently imputed. Potential interaction partners were identified by pairwise comparison with control samples, and functional differences between FL and PR were highlighted by pairwise comparisons between FL and PR. Data can be visualized in different ways, for example, a volcano plot. **(B)** WB results of initial Virotrap screens for the detection of expression and recruitment into the VLPs of GAG–bait fusion proteins. HEK293T cells were transfected with GAG–bait constructs. Additional co-transfection of VSV-G/FLAG-VSV-G expression constructs allowed VLP purification, which was followed by direct on-bead lysis and analysis by Western blotting using anti-GAG (bait expression levels and particles) and anti-α-actin antibodies (as a loading control for cell lysates). Results of VLPs and producer cell lysates are shown. For GAG-eDHFR and GAG-PAIP1 (both FL and PR), a clear band can be detected at the desired molecular weight in both VLPs and cells. For RARS, all constructs seem well expressed in the cells, but only RARS FL is recruited into VLPs. CAST is found only weakly expressed in cells and is not pulled into VLPs. Uncropped gel images and molecular weight markers are shown in Fig S2B.

---

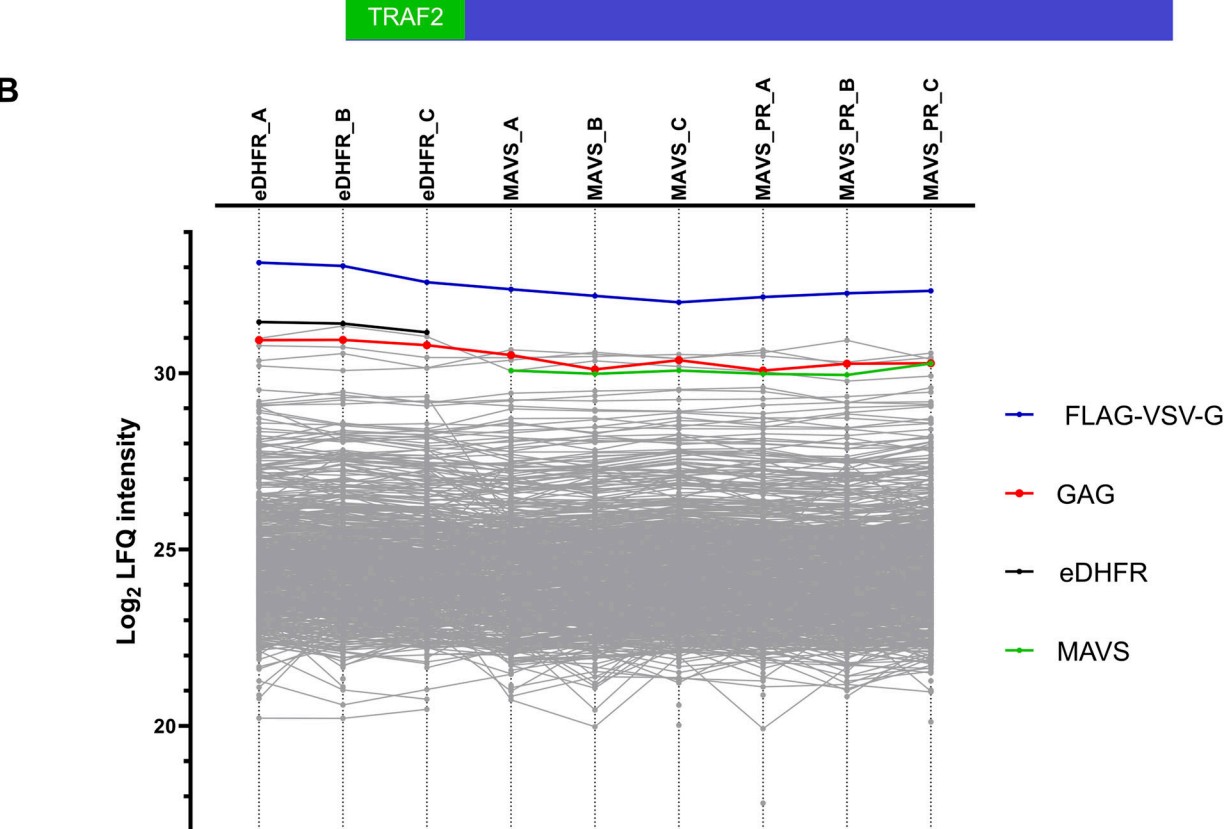

**Figure 5. Comparisons of bait intensities.**
**(A)** Schematic representation of the differences in domains for full-length MAVS and its N-terminal proteoform. **(B)** Profile plot showing the log$_2$-transformed LFQ intensities for the different identified proteins in the MAVS Virotrap experiment. LFQ intensities for every replicate are shown before imputation of missing values for statistical analysis.

On the contrary, the N-terminal proteoform (missing amino acids 1–211) of the eukaryotic initiation factor 4A-I (EIF4A1) loses several known interactions (see Fig 7D). In the pairwise comparison between FL and PR, we found that the interaction with eight candidate eukaryotic initiation factors (EIF4A3, EIF4E, EIF4G3, EIF4B, EIF4G2, EIF4A2, EIF4H, and EIF4G1) is specific for the FL, whereas on the side of the EIF4A1 proteoform, we identified several proteins involved in proteasome-mediated protein degradation as possible interactors.

### Y2H screens to validate the interaction profile of N-terminal proteoforms

To support our previous findings on the interactome of N-terminal proteoforms, we performed a yeast two-hybrid (Y2H) screen as reported in References (8, 44). These screens were performed such that all baits (both canonical protein and N-terminal proteoform) were fused to the Gal4 DNA-binding domain (DB) and tested against proteins encoded by the hORFeome v9.1 collection containing 17,408 ORF clones fused to the Gal4 activation domain (AD). After first-pass screening, each bait was pairwise tested for interaction with all the candidate partners identified for any proteoform of that gene. Pairs showing a positive result were subjected to a pairwise retest, and PCR products amplified from the final positive pairs were sequenced to confirm the identity of clones encoding each interacting protein. This resulted in the identification of 39 high-confident binary PPIs (Table S8). Note that not for all baits, PPIs are reported, which is due to the auto-activation of the reporter gene for some baits, whereas for other baits, no interactions were found or did not result in a positive pair after pairwise tests.

Of the 39 high-confident binary PPIs reported with Y2H, 11 (28.2%) are also reported as candidate interaction partners by Virotrap (Table S8). As in general, the overlap between different PPI methods is not high (45, 46, 47), this relative high overlap shows the quality of both datasets. As an example, for EIF4A1, the specific interaction of

**Table 3. List of MAVS candidate interaction partners.**

| MAVS FL | MAVS PR | Gene name | Protein accession | Protein name | PPI databases |
|---|---|---|---|---|---|
| Yes | | ACLY | P53396 | ATP-citrate synthase | |
| Yes | | EEF2 | P13639 | Elongation factor 2 | |
| Yes | | PKM | P14618 | Pyruvate kinase PKM | |
| Yes | | TRAF2 | Q12933 | TNF receptor–associated factor 2 | Yes |
| Yes | Yes | ADK | P55263 | Adenosine kinase | |
| Yes | Yes | AHCY | P23526 | Adenosylhomocysteinase | |
| Yes | Yes | ASNA1 | O43681 | ATPase ASNA1 | |
| Yes | Yes | ASS1 | P00966 | Argininosuccinate synthase | |
| Yes | Yes | BAG6 | P46379 | Large proline-rich protein BAG6 | Yes |
| Yes | Yes | BROX | Q5VW32 | BRO1 domain–containing protein BROX | |
| Yes | Yes | CHMP1A | Q9HD42 | Charged multivesicular body protein 1a | |
| Yes | Yes | CHMP1B | Q7LBR1 | Charged multivesicular body protein 1b | |
| Yes | Yes | CHMP2A | O43633 | Charged multivesicular body protein 2a | |
| Yes | Yes | CHMP4B | Q9H444 | Charged multivesicular body protein 4b | |
| Yes | Yes | DYNC1H1 | Q14204 | Cytoplasmic dynein 1 heavy chain 1 | |
| Yes | Yes | EPRS | P07814 | Bifunctional glutamate/proline–tRNA ligase | |
| Yes | Yes | FERMT3 | Q86UX7 | Fermitin family homolog 3 | |
| Yes | Yes | FLNA | P21333 | Filamin-A | |
| Yes | Yes | GAPDH | P04406 | Glyceraldehyde-3-phosphate dehydrogenase | |
| Yes | Yes | GET4 | Q7L5D6 | Golgi-to-ER traffic protein 4 homolog | |
| Yes | Yes | HLA-C | P10321 | HLA class I histocompatibility antigen, Cw-7 alpha chain | |
| Yes | Yes | HSPB1 | P04792 | Heat shock protein $\beta$-1 | |
| Yes | Yes | HYOU1 | Q9Y4L1 | Hypoxia up-regulated protein 1 | |
| Yes | Yes | IFIT1 | P09914 | Interferon-induced protein with tetratricopeptide repeats 1 | Yes |
| Yes | Yes | IFNB1 | P01574 | Interferon beta | Yes |
| Yes | Yes | ILK | Q13418 | Integrin-linked protein kinase | |
| Yes | Yes | IST1 | P53990 | IST1 homolog | |
| Yes | Yes | MAPRE2 | Q15555 | Microtubule-associated protein RP/EB family member 2 | |
| Yes | Yes | MAVS | Q7Z434 | Mitochondrial antiviral-signaling protein | |
| Yes | Yes | MITD1 | Q8WV92 | MIT domain–containing protein 1 | |
| Yes | Yes | PRPF19 | Q9UMS4 | Pre-mRNA–processing factor 19 | |
| Yes | Yes | RAD23B | P54727 | UV excision repair protein RAD23 homolog B | |
| Yes | Yes | RBBP7 | Q16576 | Histone-binding protein RBBP7 | |
| Yes | Yes | SAR1A | Q9NR31 | GTP-binding protein SAR1a | |
| Yes | Yes | SEPT2 | Q15019 | Septin-2 | |
| Yes | Yes | SGTA | O43765 | Small glutamine-rich tetratricopeptide repeat–containing protein alpha | |
| Yes | Yes | TLN1 | Q9Y490 | Talin-1 | |
| Yes | Yes | UBL4A | P11441 | Ubiquitin-like protein 4A | |
| Yes | Yes | UBQLN1 | Q9UMX0 | Ubiquilin-1 | |
| Yes | Yes | UBQLN4 | Q9NRR5 | Ubiquilin-4 | |
| Yes | Yes | VPS4A | Q9UN37 | Vacuolar protein sorting–associated protein 4A | |
| Yes | Yes | VTA1 | Q9NP79 | Vacuolar protein sorting–associated protein VTA1 homolog | |

**Table 3.  Continued**

| MAVS FL | MAVS PR | Gene name | Protein accession | Protein name | PPI databases |
|---------|---------|-----------|-------------------|--------------|---------------|
| Yes | Yes | WDR1 | O75083 | WD repeat–containing protein 1 | |
| | Yes | ACTL6A | O96019 | Actin-like protein 6A | |
| | Yes | CAD | P27708; P31327 | CAD protein | |
| | Yes | DPYSL2 | Q16555 | Dihydropyrimidinase-related protein 2 | |
| | Yes | PGM1 | P36871 | Phosphoglucomutase-1 | |
| | Yes | PKRKACB | P22694 | cAMP-dependent protein kinase catalytic subunit beta | |
| | Yes | PLK1 | P53350 | Serine/threonine protein kinase PLK1 | |
| | Yes | PRKDC | P78527 | DNA-dependent protein kinase catalytic subunit | |
| | Yes | PYGL | P06737 | Glycogen phosphorylase, liver form | |
| | Yes | SEPT7 | Q16181 | Septin-7 | |

MAVS FL and PR Virotrap experiments were performed in triplicate, and data were compared with eDHFR control samples. The first two columns indicate whether a protein was identified as a candidate MAVS FL or PR interaction partner, and the last column indicates whether a protein is a known MAVS interaction partner as listed in the BioGRID, STRING, and/or IntAct databases.

the canonical protein (and not the PR) with PDCD4 (a well-known interaction partner) is supported by both Virotrap and Y2H.

### Studying selected differences between proteoform interactomes by affinity purification–mass spectrometry (AP-MS)

We selected three baits, MAVS, EIF4A1, and PAIP1, for further validation by AP-MS. Both FL–bait–FLAG and PR–bait–FLAG fusion constructs were generated in which FLAG is fused to the C-terminus of the bait to avoid steric hindrance of the tag on the bait's N-terminus. Four biological repeats of pull-down experiments using FLAG-tagged baits and an eDHFR-FLAG control were performed. Quantitative mass spectrometry was used to quantify the interaction partners of all proteoforms. In total, 1,903 proteins were identified over all experiments (Table S9). Pairwise contrasts between control–bait samples and between proteoform samples were selected at a Benjamini–Hochberg adjusted $P$-value (FDR) ≤ 0.05 (Table S9, second tab). Such tests between eDHFR control samples and baits resulted in 14 candidate interaction partners for MAVS FL, whereas for MAVS PR, only the bait was found as being significant (Tables 4 and S9, second tab).

We compared our list of potential interaction partners with known interactors listed in BioGRID (48), STRING (30), and IntAct (49), and with candidate interactors identified by Virotrap. Six of 14 candidate interaction partners were reported in at least one of the consulted databases, whereas just two candidates were also reported by Virotrap. The rather high overlap with known interaction partners points to the quality of the AP-MS data.

When applying the same selection criteria as used in the pairwise tests between Virotrap data for FL and PR, the pairwise comparison between the MAVS FL and PR interactomes reveals several candidate interaction partners that are enriched in the MAVS FL interactome (Fig 7E and Table S9, tab 4). In fact, our AP-MS data support our Virotrap findings that TRAF2 is enriched in MAVS FL interactomes.

For PAIP1, we identified several candidate interactors of both FL and PR, nine and 12, respectively, of which seven are known interaction partners and two were also reported as interaction partners by Virotrap (Table S9). However, none of the members of

the 4EHP-GYF2 complex were found as candidate interaction partners. We could thus not support these specific Virotrap findings by AP-MS. We hypothesize that this could be due to the differences in the PPI techniques as Virotrap allows the detection of weaker and transient interactions because of the avidity effect of multiple bait copies lining the inside of VLPs. A pairwise comparison between PAIP1 FL and PR resulted in one significant protein, UBE2T, which is reported to be enriched in proteoform samples. However, this protein was not listed as a candidate interaction partner before.

For EIF4A1 FL, only two candidate interactors were found: EIF4A2 (known interaction partner, also reported by Virotrap) and IFNA2. For the corresponding PR, no candidate interactors could be identified. These two proteins were also reported as significant in the comparison between the FL and PR, with the proteins found to be specific for the FL protein. Our AP-MS data thus support that the interaction of EIF4A1 with EIF4A2 is lost for the N-terminal proteoform, as also found by Virotrap.

Based on our Virotrap and AP-MS interactomics data for both the canonical protein and N-terminal proteoform of MAVS and cross-checked with known interactors listed in BioGRID (48), STRING (30), and IntAct (49), we generated a PPI network of MAVS (Fig 8), showing all identified candidate interaction partners. Each edge represents an identified interaction between the bait (either MAVS FL or PR, red nodes) and the prey protein. In total, we identified 65 proteins and 115 interactions. Most of the interaction partners (38) are shared between the canonical protein and the N-terminal proteoform (clustered in the middle). However, for both MAVS FL and PR, we also report a set of unique interaction partners (clustered on the left and right sides). For the canonical protein, we found 16 preys that solely interact with the canonical protein, whereas for the proteoform, we report nine unique interaction partners. From all identified potential interaction partners, nine are known interaction partners listed in at least one of the three consulted PPI databases. This shows that both Virotrap and AP-MS identified known interaction partners, and besides that both approaches also reported several novel potential interaction partners, many of which are only identified with one of the two methods. This again

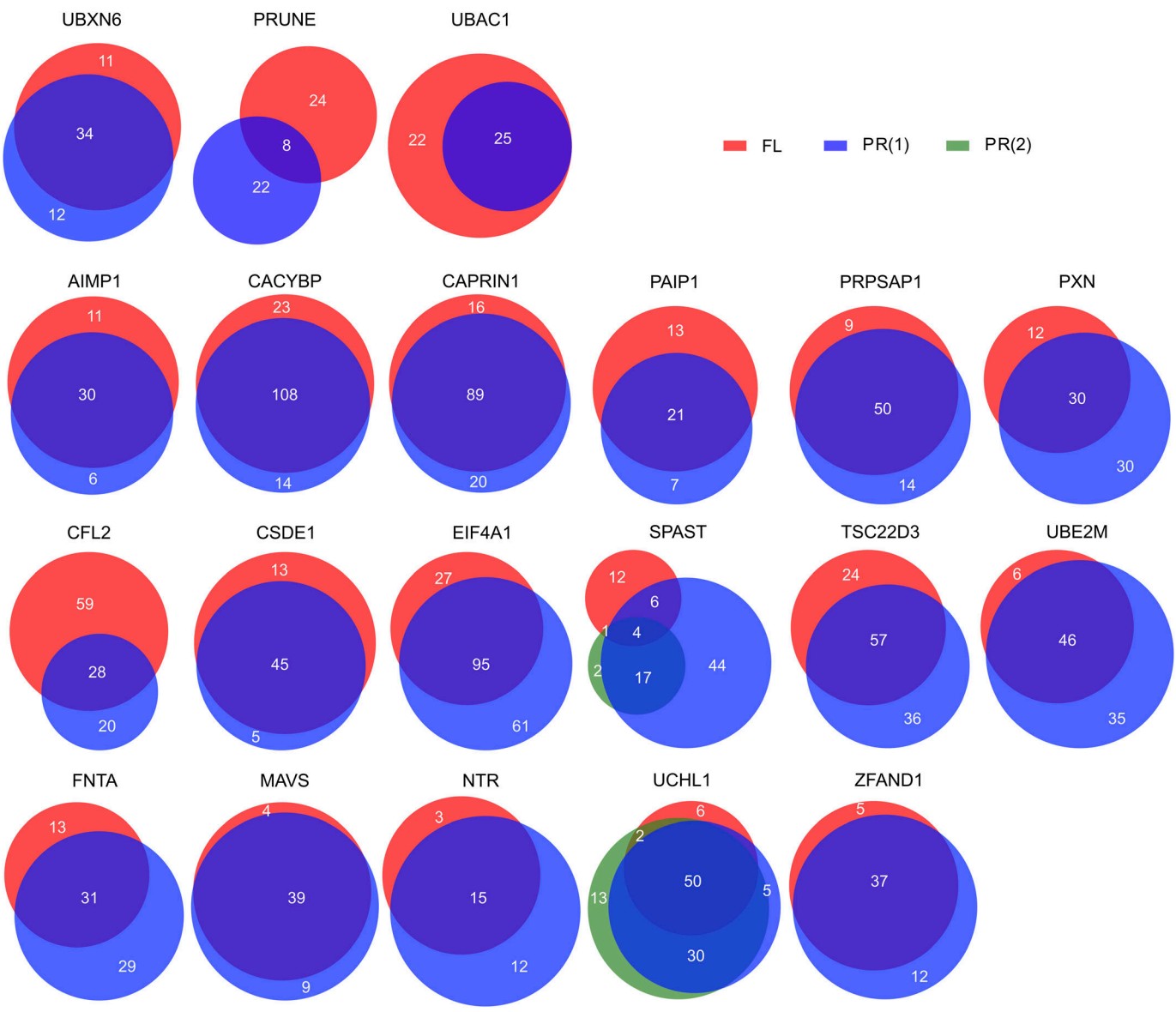

**Figure 6. Overview of the overlap between the candidate interaction partners identified for the FL and the PR baits.**
The three baits on the top row show three different scenarios of differences between the interactome identified for the canonical protein and the N-terminal proteoform. The overlap between FL and PR of all other baits is shown.

demonstrates the uniqueness of each PPI method and the difficulties often encountered when validating specific interactions. In addition, all unique or specific interaction partners reported for MAVS PR are not reported in either BioGRID, STRING, or IntAct. This could hint at an alternative function of the proteoform, which is unrelated to the function of the FL protein.

## Discussion

Several studies have reported on the chemical diversity of N-terminal proteoforms (9, 14, 17, 18, 20, 21, 22, 23, 24, 25, 50, 51); however,

their functional diversity has not been investigated on a large scale. We here used positional proteomics to build a map of N-terminal proteoforms in the HEK293T cellular cytosol and identified 1,044 N-terminal proteoforms (Table 1). From this map, 20 pairs of N-terminal proteoforms and their canonical proteins were selected to map their PPIs (Fig 4 and Table 2). Interaction networks of all proteins were generated, as their quality was validated by checking the overlap with known interactors listed in BioGRID (50), STRING (51), and IntAct (52) (Tables S6, S7, and S9). On average, N-terminal proteoforms share >60% of their interactions with their canonical counterparts (Fig 5D and Table S5, second tab). However, for all studied pairs, we could report interactome differences, suggesting functional divergence between proteoforms, and noticed both the

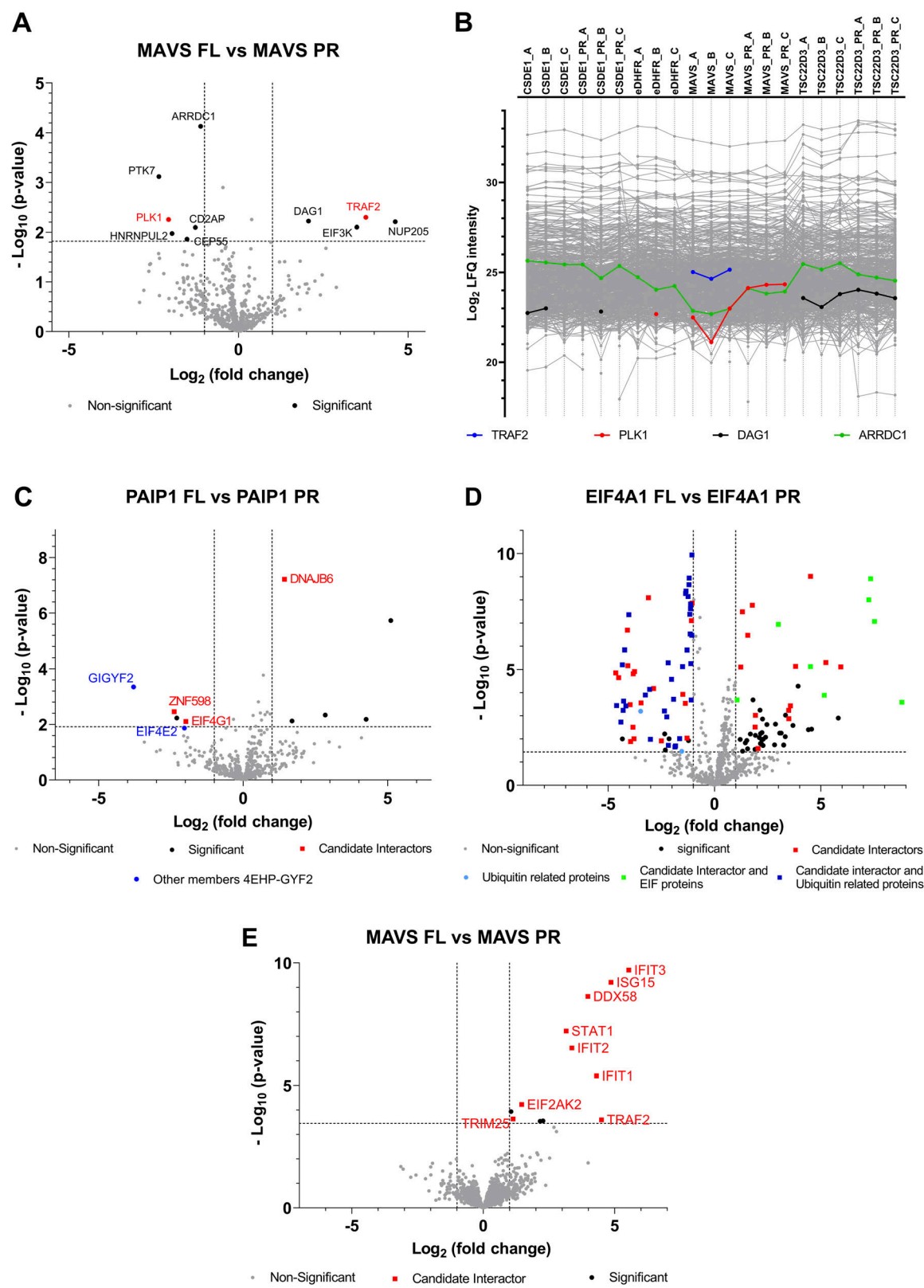

**Figure 7. Interactomics results of selected baits with Virotrap and AP-MS.**
**(A)** Volcano plot showing the candidate interaction partners that differ between MAVS FL (right) and PR (left) as identified with Virotrap. The x-axis shows the $\log_2$ fold change of the interactors' intensities in the MAVS FL samples (right) relative to the MAVS PR samples (left). The y-axis shows the $-\log_{10}$ $P$-value of the adjusted $P$-value. Significant differences were defined through pairwise comparisons between MAVS FL and PR samples, with a FDR ≤ 0.05 (the corresponding $-\log_{10}$ of the adjusted $P$-value

loss of interactions and the engagement of proteoforms in novel interactions (Fig 8).

Upon analyzing data of all pairs, some proteins were found as candidate interaction partners for several baits, which have highly different functions. Therefore, similar as described in References (35, 52), one could think of challenging potential interactors with a list of all identified interactors in different, unrelated Virotrap experiments. In this way, proteins found in a large fraction of the whole dataset may be proteins present in VLPs irrespective of the bait and could be removed (53). We compiled a list of all identified proteins (and their frequency) for all FL-PR (and PR1-PR2) comparisons and an alike list of all candidate interaction partners found in the bait–control comparisons (Table S10). Of note, CHMP4B and RAD23B are listed as candidate interactors of 17 baits and are thus more likely not true interactors. Among the frequently identified candidate interaction partners, we also find GAG (in 11 of 20 comparisons). This is not unexpected as we reported differences in intensities between eDHFR-GAG levels and bait–GAG levels (Fig S3). As a consequence, we also found several known GAG interaction partners among the frequently identified candidate interactors such as BROX (reported for 14 baits), SLC3A2 and TSG101 (12 baits), and PDCD6IP (11 baits). For the comparisons between proteoforms (24 in total), we identified a total of 416 proteins that differ between at least one pair of FL/PR. MLF2 is reported in 11 comparisons. These are unlikely to point to specific differences between FL and PR interactomes, but are again more likely contaminants. Many proteasome-related proteins were frequently identified, as shown above for EIF4A1 (Fig 7D), which indicates that one should also consider the possibility that such proteins might not be true interactors of a bait but rather inherently present in Virotrap VLPs. Of note, using AP-MS, we did not identify such proteins as candidate interaction partners for any of our baits, which strengthens our hypothesis that such proteins might be contaminants inherent to the Virotrap method.

Validation of Virotrap data was done by Y2H on all 20 pairs of proteoforms. Y2H identified 39 PPIs providing extra evidence supporting our Virotrap data. The lower number of PPIs found by Y2H could partially be explained by the technology's limitations and the considered approach, as baits were only tested in one setup (bait coupled to AD and tested for interaction with prey coupled to DB, and not in other setups such as bait coupled C-terminally to AD or bait coupled to DB). Therefore, we found that Y2H did not work for most of the baits, also likely, because of auto-activation or failure of nuclear localization of bait and/or prey (8, 54). Nevertheless, we were able to validate the interaction of PDCD4 with the full-length EIF4A1 protein, reported by Virotrap.

For MAVS, PAIP1, and EIF4A1, AP-MS experiments were performed to validate interactome differences between the FL and PR reported by Virotrap. For the N-terminal proteoform of MAVS, we could

validate the loss of interaction with TRAF2, whereas for the N-terminal proteoform of EIF4A1, the loss of interaction with EIF4A2 was validated. Note that none of the other EIF proteins that Virotrap reported as specific interaction partners of the canonical EIF4A1 protein nor the interaction of the N-terminal EIF4A1 proteoform with proteasome- and ubiquitin-related proteins was validated by AP-MS. For PAIP1, we were not able to validate the interaction of its N-terminal proteoform with members of the 4EHP-GYF2 complex. The differences between the candidate interactors reported with Virotrap or AP-MS might be inherently caused by the methods used. In AP-MS, samples are lysed, which likely leads to mixing of proteins that are not in the same localization, leading to false-positive PPIs. In addition, several washing steps are needed, which break weaker and transient interactions. Virotrap, on the contrary, traps the protein complexes in VLPs, protecting them during purification. Moreover, the GAG grid-like structure creates an avidity effect for the bait, and thus in general, Virotrap allows to detect weaker and transient interactions (35, 55). As both methods detected known PPIs, one cannot question the quality of both datasets. The fact that both methods also identified partially different interactomes for FL and PR baits proves our working hypothesis and, moreover, once again shows the complementarity of different PPI methods.

Other interesting differences were also evident from the AP-MS data (Fig 7E). For instance, it was reported that IFIT3 interacts with MAVS through the N-terminus of IFIT3 (56). However, as this interaction is enriched in the MAVS FL interactome, it hints at the fact that the N-terminus of MAVS is also important for this interaction. Besides IFIT3, several other (related) proteins are also listed to differ between FL and PR (being enriched as FL binders), and these proteins include IFIT1, IFIT2, IFIT5, ISG15, and STAT1. IFIT3 is known to interact with these proteins (see the interaction network of IFIT3 as listed by STRING (30)), and IFIT1, IFIT2, and IFIT3 form a protein complex (57). Upon viral infection, IFIT3 expression is up-regulated, which limits the replication of RNA viruses (by direct inhibition of translation); however, IFIT3 has also an indirect antiviral effect through its interaction with MAVS. When interferon (IFN) signaling is activated, IFIT3 induces the expression of IFIT1 and IFIT2, which form a complex and stimulate TBK1 phosphorylation through MAVS. This leads to IRF3 phosphorylation and IFNβ gene expression, resulting in the up-regulation of several ISGs and the phosphorylation of STAT1 (58), activating canonical IFN signaling. Our interactome data let us hypothesize that IFIT3 interacts with MAVS through the N-terminus of MAVS and that several other proteins listed as significant in the comparison between the MAVS FL and PR interactomes are due to their involvement in the IFN signaling pathway in which the interaction between IFIT3 and MAVS plays a central role. It has been reported that the N-terminal truncated proteoform of MAVS we have identified becomes the dominant expressed MAVS proteoform upon viral infection. Moreover, this

---

to a FDR of 0.05 is shown as cutoff) and a fold change larger than |1| (cutoffs also shown on a volcano plot). Proteins retained after filtering for candidate interaction partners are highlighted in red. **(B)** Profile plot showing the LFQ intensities of all identified proteins using Virotrap. LFQ intensities of proteins quantified in every biological replicate are shown before imputation of missing values. Profiles of selected proteins reported as significantly different between MAVS FL and PR interactomes are highlighted. **(A, C)** Volcano plot, similar as described in (A) for PAIP1 FL and PR interactomes identified with Virotrap. Candidate interaction partners that significantly differ between PAIP1 FL and PR (FDR 0.05 and fold change > |1| log₂ are shown in red). Other members, not listed as candidate interactors, of the 4EHP-GYF2 complex are shown in blue. **(A, D)** Volcano plot similar as described in (A) for EIF4A1 FL and PR interactomes as identified with Virotrap. **(A, E)** Volcano plot similar as described in (A) for MAVS FL and PR interactomes as identified with AP-MS.

**Table 4.** List of MAVS candidate interaction partners identified by AP-MS.

| MAVS FL | MAVS PR | Gene name | Protein accession | Protein name | PPI databases | Virotrap |
|---------|---------|-----------|-------------------|--------------|---------------|----------|
| Yes | | DDX58 | O95786 | Probable ATP-dependent RNA helicase DDX58 | Yes | |
| Yes | | EIF2AK2 | P19525 | Interferon-induced, double-stranded RNA-activated protein kinase | Yes | |
| Yes | | HLA-C | P10321 | HLA class I histocompatibility antigen, Cw-7 alpha chain | | Yes |
| Yes | | HMGCS1 | Q01581 | Hydroxymethylglutaryl-CoA synthase, cytoplasmic | | |
| Yes | | IFIT1 | P09914 | Interferon-induced protein with tetratricopeptide repeats 1 | Yes | Yes |
| Yes | | IFIT2 | P09913 | Interferon-induced protein with tetratricopeptide repeats 2 | | |
| Yes | | IFIT3 | O14879 | Interferon-induced protein with tetratricopeptide repeats 3 | Yes | |
| Yes | | ISG15 | P05161 | Ubiquitin-like protein ISG15 | | |
| Yes | | KIF2C | Q99661 | Kinesin-like protein KIF2C | | |
| Yes | | SLC39A7 | Q92504 | Zinc transporter SLC39A7 | | |
| Yes | | STAT1 | P42224 | Signal transducer and activator of transcription 1-alpha/beta | Yes | |
| Yes | | TPRKB | Q9Y3C4 | EKC/KEOPS complex subunit TPRKB | | |
| Yes | | TRIM25 | Q14258 | E3 ubiquitin/ISG15 ligase TRIM25 | Yes | |
| Yes | | UBA52; RPS27A; UBB; UBC | P62987 | Ubiquitin–60S ribosomal protein L40 | | |

MAVS AP-MS experiments, with FL or PR bait, were performed with four replicates and were challenged against eDHFR control samples. The first two columns indicate whether a protein was identified as a candidate MAVS interaction partner for the FL or PR. The last but one column shows whether a protein is a known MAVS interaction partner listed in BioGRID, STRING, and/or IntAct. The last column indicates the overlap with the candidate interaction partners identified by Virotrap.

N-terminal proteoform reduces antiviral responses by interfering with interferon production and STAT1 phosphorylation (59).

In this study, Virotrap was used to map PPIs. However, as this method currently works best in HEK293T cells on cytosolic proteins, this limits the N-terminal proteoforms that can be studied. The localization of various N-terminal proteoforms is known to be affected (12, 42, 60, 61, 62, 63, 64, 65, 66, 67), making it possible that N-terminal proteoforms exert similar or different functions at different localizations. However, this interesting possibility could not be investigated as Virotrap is mainly restricted toward cytosolic proteins, but other approaches such as proximity labeling could be used in the future for this type of analysis. Moreover, in Virotrap, the N-terminus of the bait is fused to the C-terminus of GAG (the N-terminus of GAG is essential for its coupling to the plasma membrane). The N-terminus of the bait is thus not free, and the neighborhood of GAG might sterically hinder prey proteins from interacting with the N-terminus of the bait. A decoupled variant of Virotrap, where GAG and bait are free and only coupled together upon addition of a dimerizer, could avoid this issue.

We here attempted a more global analysis of the interaction profiles of N-terminal proteoforms and their canonical counterparts. However, as Virotrap and other MS-based PPI methods are labor-intensive, only a limited, yet well-selected, set of pairs could be analyzed. Funneling all identified N-terminal proteoforms to a manageable set for Virotrap-based interactome analysis was based on several selection criteria that could have introduced biases.

First, as mentioned, we focused on cytosolic proteoforms. Second, the identified peptides were stringently filtered to only select N-terminal peptides confidently originating from translation events, thus ignoring N-terminal proteoforms originating from protein processing, which further reduced the N-terminal proteoforms that could be considered for further analysis. The fraction of N-terminal proteoforms among all identified peptides is much smaller than expected from the composition of our database (see Fig 3A in Reference (36)). From 22,003 UniProt isoforms, we identified 89, of which 83 were highly confident identifications. In analogy, from 60,661 proteoforms predicted from Ribo-seq data, we could only identify 80, including 73 with high confidence. Similar disproportions have been reported by (N-terminal) riboproteogenomics studies (5, 68, 69, 70) and, in our opinion, highlight the need for validation of candidate proteoforms derived from Ribo-seq analysis by alternative means. After initial filtering, a second orthogonal selection based on biological information was made to retain the potentially most interesting pairs. From this list, a final selection was made based on criteria to prioritize candidates suited for Virotrap, which meant prioritizing proteoforms with a substantial difference at their N-termini, non-structural proteins, and proteoforms with losses or gains of protein domains or motifs. This likely also introduced a bias in our results as these criteria might increase our chances of identifying differences between the interactomes of the studied proteoforms. As such,

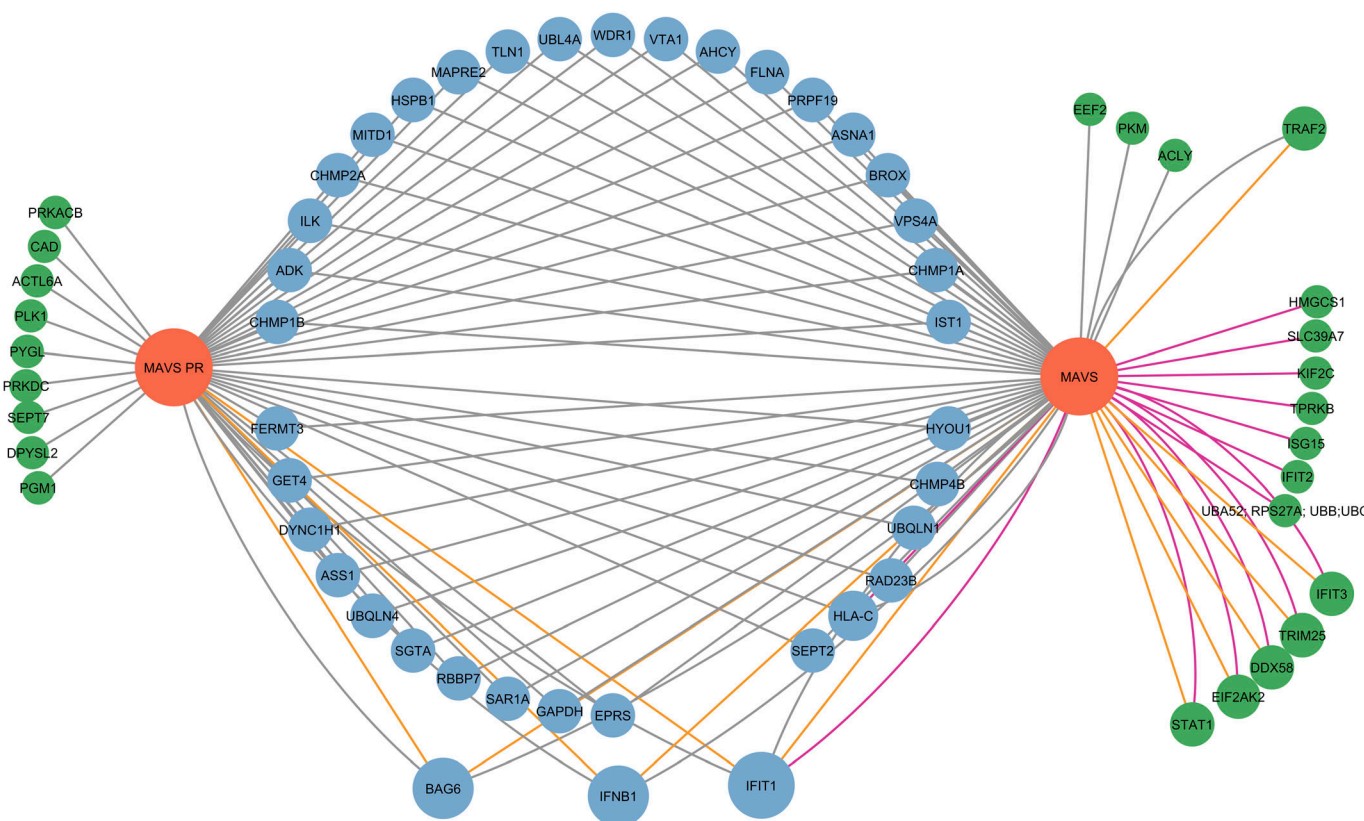

**Figure 8.   Protein–protein interaction network for proteoforms of MAVS.**
This network was generated by Cytoscape (version 3.9.1). Bait proteins (MAVS FL and MAVS PR) are shown in red. All nodes represent interaction partners identified by either Virotrap or AP-MS. Gray edges indicate interactions identified by Virotrap, whereas purple edges indicate interactions reported by AP-MS. Orange edges represent that the interaction, reported by either Virotrap or AP-MS, is supported by public PPI databases (either BioGRID, STRING, or IntAct). The size of the nodes is related to how many edges are linked to this node, indicative of higher confidence for a given prey. In total, the network contains 65 nodes (proteins) and 115 edges (interactions). The interactions shared between the canonical protein and the N-terminal proteoform are clustered in the middle (blue nodes), whereas the proteins on the left and right sides represent unique interaction partners of the N-terminal proteoform and canonical protein, respectively (green nodes).

it will likely be error-prone to translate our results toward general conclusions about N-terminal proteoforms.

Our inspection of publicly available tissue data from different tissues (40) not only showed that N-terminal proteoforms are expressed in different tissues, increasing their biological relevance, but also showed that some proteoforms are expressed in a tissue-specific manner. We only mapped PPIs in HEK293T cells, but it seems interesting to study PPIs of N-terminal proteoforms in these tissues (or conditions, e.g., stress conditions) where these are normally expressed in.

In summary, we report confident maps of PPIs of 20 pairs of N-terminal proteoform(s) and their canonical proteins that can be explored further by the research community. Overall, our results show that N-terminal proteoforms expand the functional diversity of a proteome, which highlights the importance of considering proteoforms when studying the function of a given protein. Moving forward, studies mapping the proteoforms and their interactors in different tissues and conditions would help our understanding of the functional complexity of the proteome and our understanding of disease pathologies. However, performing such studies in a more systematic way remains challenging and labor-intensive. Further improvements in PPI methods that would allow large-scale screens seem necessary.

# Materials and Methods

### Cell culture

Human embryonic kidney cells (HEK293T cells) were cultured at 37°C and 8% $CO_2$ in DMEM supplemented with 10% FBS (unless specified otherwise).

### Cytosol extraction

Cytosolic extracts were prepared from 2.5 × 10$^7$ HEK293T cells as described in Reference (71). Cell pellets were washed with ice-cold DPBS (cat. No. 14190250; Thermo Fisher Scientific), re-suspended in 1.25 ml of cell-free system buffer (220 mM mannitol, 170 mM sucrose, 5 mM NaCl, 5 mM $MgCl_2$, 10 mM Hepes, pH 7.5, 2.5 mM $KH_2PO_4$, 0.02% digitonin, and cOmplete protease inhibitor cocktail [cat. No. 4693132001; Roche]), and kept on ice for 2 min. Lysates were cleared by centrifugation at 14,000g for 15 min at 4°C. The supernatants,

containing the cytosolic extracts, were collected. The pellets and a HEK293T total lysate serving as a control were re-suspended in 1 ml of RIPA buffer (50 mM Tris–HCl, pH 7.4, 150 mM NaCl, 1% NP-40, 0.5% sodium deoxycholate, 0.1% SDS, and cOmplete protease inhibitor cocktail) followed by three freeze–thaw cycles. Lysates were cleared by centrifugation at 16,000$g$ for 15 min at room temperature. 200 $\mu$l of each sample was used for WB analysis, whereas the rest of the sample was used for N-terminal peptide enrichment by COFRADIC.

### WB analysis

Proteins were denatured in XT sample buffer (cat. No. 1610791; Bio-Rad) and XT reducing agent (cat. No. 1610792; Bio-Rad), heated at 99°C for 10 min, and centrifuged for 5 min at 16,000$g$ (room temperature). 25 $\mu$g of each protein mixture was separated by SDS–PAGE (polyacrylamide gel electrophoresis) on a Criterion XT 4–12% Bis-Tris gel (cat. No. 3450124; Bio-Rad Laboratories). Proteins were transferred to a PVDF membrane (cat. No. IPFL00010; Merck Millipore) after which the membrane was blocked using Odyssey blocking buffer (PBS) (cat. No. 927-4000; LI-COR) diluted once with TBS-T (TBS supplemented with 0.1% Tween-20). Immunoblots were incubated overnight with primary antibodies against GAPDH (ab8245; Abcam), HSP60 (sc-13115; Santa Cruz), lamin B (sc-374015; Santa Cruz), ribophorin I (sc-12164; Santa Cruz), or $\gamma$-tubulin (MA1-850; Thermo Fisher Scientific) in Odyssey blocking buffer (PBS) diluted once with TBS-T. Blots were washed four times with TBS-T and incubated with fluorescently labeled secondary antibodies (IRDye 800CW Goat Anti-Mouse IgG polyclonal 0.5 mg from cat. No 926-32210; LI-COR and IRDye 800CW Donkey Anti-Goat IgG polyclonal 0.5 mg from cat. No. 926-32214; LI-COR) in Odyssey blocking buffer diluted once with TBS-T for 1 h. After three washes with TBS-T and an additional wash in TBS, immunoblots were imaged using the Odyssey infrared imaging system (LI-COR).

### N-terminal COFRADIC

N-terminal peptides were enriched by COFRADIC as described previously (72), however, without the pyroglutamate removal and SCX steps. In the following, we only mention the main differences with the published protocol. In brief, 1 mg of cytosolic proteins was used. As digitonin was used for cytosolic extraction in combination with the RIPA lysis buffer, which interferes with LC-MS/MS analysis, the samples were cleaned up using Pierce Detergent removal spin columns (cat. No. 87777; Thermo Fisher Scientific) according to the manufacturer's instructions. Then, guanidinium hydrochloride was added to a final concentration of 4 M before proteins were reduced (with 15 mM f.c. [final concentration] of TCEP) and alkylated (with 30 mM f.c. of iodoacetamide) for 15 min at 37°C. To be able to distinguish between in vivo Nt-acetylation events and free primary amines, all primary protein amines were blocked using stable isotope–encoded acetate, that is, an NHS ester of $^{13}C_1D_3$-acetate. The acetylation reaction was allowed to proceed for 1 h at 30°C and was repeated once. Before digestion, the samples were desalted on a NAP-10 column in 50 mM freshly prepared ammonium bicarbonate, pH 7.8. Samples were digested with either trypsin in a trypsin/substrate ratio of 1/50 (w/w) (cat. No. V5111; Promega) and incubated overnight at 37°C, chymotrypsin in a chymotrypsin/

substrate ratio of 1/20 (w/w) (cat. No. V1061; Promega) and incubated overnight at 25°C, or endoproteinase GluC in a GluC/substrate ratio of 1/20 (w/w) (cat. No.V1651; Promega) and incubated overnight at 37°C. After vacuum drying, the samples were re-dissolved in 80 $\mu$l loading solvent A (2% acetonitrile [ACN] and 0.1% TFA in ddH$_2$O) before isolating N-terminal peptides by two subsequent RP-HPLC fractionations with a TNBS reaction in between.

### LC-MS/MS analysis of N-terminal peptides and peptide identification

LC-MS/MS analysis was performed similarly as reported before (72). Each COFRADIC fraction was re-solubilized in 20 $\mu$l loading solvent A, and half of each fraction was injected for LC-MS/MS analysis on an UltiMate 3000 RSLCnano system in-line connected to an Orbitrap Fusion Lumos mass spectrometer (Thermo Fisher Scientific). Trapping was performed at 10 $\mu$l/min for 4 min in loading solvent A on a 20-mm trapping column (made in-house, 100 $\mu$m internal diameter [I.D.], 5-$\mu$m beads, C18 Reprosil-HD, Dr. Maisch, Germany). The peptides were separated on a 200-cm $\mu$PAC column (C18-endcapped functionality, 300-$\mu$m-wide channels, 5-$\mu$m porous-shell pillars, inter-pillar distance of 2.5 $\mu$m, and a depth of 20 $\mu$m; PharmaFluidics). The column was kept at a constant temperature of 50°C. Peptides were eluted by a linear gradient reaching 33% MS solvent B (0.1% formic acid [FA] in water/ACN [2:8, vol/vol]) after 42 min, 55% MS solvent B after 58 min, and 99% MS solvent B at 60 min, followed by a 10-min wash at 99% MS solvent B and re-equilibration with MS solvent A (0.1% FA in water). For the first 15 min, the flow rate was set to 750 nl/min after which it was kept constant at 300 nl/min.

The mass spectrometer was operated in a data-dependent mode, automatically switching between MS and MS/MS acquisition. Full-scan MS spectra (300–1,500 m/z) were acquired in 3-s acquisition cycles at a resolution of 120,000 in the Orbitrap analyzer after accumulation to a target AGC value of 200,000 with a maximum injection time of 250 ms. The precursor ions were filtered for charge states (2–7 required), dynamic range (60 s; ± 10 ppm window), and intensity (minimal intensity of 5E3). The precursor ions were selected in the ion routing multipole with an isolation window of 1.6 D, accumulated to an AGC target of 10E3 or a maximum injection time of 40 ms, and activated using CID fragmentation (35% NCE). The fragments were analyzed in the Ion Trap Analyzer at a rapid scan rate.

The generated MS/MS peak lists were searched with Mascot using the Mascot Daemon interface (version 2.6.0; Matrix Science). MS data were searched twice, once being matched against a custom-built database (containing UniProt and UniProt isoform entries appended with Ribo-seq–derived protein sequences; generation and composition are described in detail in Reference (36)) and once being matched against the UniProt database (release 2019_04). The Mascot search parameters were for both searches the same and as follows: heavy acetylation of lysine side-chains (with $^{13}C_1D_3$-acetate), carbamidomethylation of cysteine, and methionine oxidation to methionine sulfoxide were set as fixed modifications. Variable modifications were acetylation of N-termini (both light and heavy because of the $^{13}C_1D_3$ label) and pyroglutamate formation of N-terminal glutamine (both at the peptide level).

The enzyme settings were as follows: endoproteinase semi-Arg-C/P (semi-Arg-C specificity with Arg-Pro cleavage allowed) allowing for two missed cleavages for the trypsin sample. For chymotrypsin and GluC, the enzyme settings were semi-Chymo and semi-GluC. For GluC, two missed cleavages were allowed, whereas for chymotrypsin, four missed cleavages were allowed. Mass tolerance was set to 10 ppm on the precursor ion and to 0.5 D on fragment ions. In addition, the C13 setting of Mascot was set to 1. Peptide charge was set to 1+, 2+, and 3+, and instrument setting was put to ESI-TRAP. Raw DAT-result files of MASCOT were further queried using ms_lims ([73]). Only peptides that were ranked first and scored above the threshold score set at 99% confidence were withheld. The FDR was estimated by searching a decoy database (a reversed version of the custom-generated database or the UniProt database), which resulted in a FDR of 0.44% for the trypsin sample, 0.14% for the chymotrypsin sample, and 0.53% for the GluC sample for the custom-built database, and FDRs of 0.72%, 0.29%, and 1.13% for the trypsin, chymotrypsin, and GluC sample, respectively, with the UniProt database.

### Generation of a catalogue of N-terminal proteoforms in HEK293T cells

From the set of identified peptides, N-terminal peptides were selected and classified as described in Reference ([36]). Different here is that data from two database searches are combined. The merging of the data from the two searches was done per protease after accession sorting, which ensures that peptides reported by different searches (or proteases) are matched to the same accession, allowing a straightforward merge. During this merge, all information was retained (e.g., all listed accessions are retained in the isoform column), and after merging, the workflow was followed as outlined in Reference ([36]) to filter for N-terminal peptides stemming from translation events.

### Generation of a cytosolic proteome map of HEK293T cells

Cytosolic extracts (in triplicate) were prepared from $10^7$ HEK293T cells as described above. The protein concentration was measured using a Bradford assay (Bio-Rad), and 500 $\mu$g of protein material was used to prepare samples for mass spectrometry analysis. Proteins were digested with endoproteinase LysC (1:100, w:w; Promega) for 4 h at 37°C and then with trypsin (1:100, w:w; Promega) overnight at 37°C. Samples were acidified to a pH < 3 by adding TFA to a final concentration of 1% (vol/vol). After 15-min incubation on ice, the samples were centrifuged for 15 min at 1,750$g$ at room temperature. The peptide-containing supernatant was further purified using SampliQ C18 100-mg columns (Agilent) according to the manufacturer's instructions. Purified peptides were dried completely by vacuum drying and re-dissolved in 200 $\mu$l loading solvent A (0.1% TFA in water/ACN [98:2, vol/vol]). The peptide concentration was measured on a Lunatic microfluidic device (Unchained Labs), and from each replicate, 50 $\mu$g of peptide material was injected for fractionation by RP-HPLC (Agilent series 1,200) connected to a Probot fractionator (LC Packings). Peptides were first loaded in solvent A on a 4-cm pre-column (made in-house, 250 $\mu$m internal diameter [ID], 5-$\mu$m C18 beads, Dr.

Maisch) for 10 min at 25 $\mu$l/min and then separated on a 15-cm analytical column (made in-house, 250 $\mu$m ID, 3-$\mu$m C18 beads, Dr. Maisch). Elution was done using a linear gradient from 100% RP-HPLC solvent A (10 mM ammonium acetate [pH 5.5] in water/ACN [98:2, vol/vol]) to 100% RP-HPLC solvent B (70% ACN and 10 mM ammonium acetate [pH 5.5]) in 100 min at a constant flow rate of 3 $\mu$l/min. Fractions were collected every minute between 20 and 85 min and pooled to generate a total of 10 samples for LC-MS/MS analysis. All 10 fractions were dried under vacuum in HPLC inserts and stored at −20°C until use.

Peptides were re-dissolved in 20 $\mu$l of 0.1% TFA and 2% ACN, and 15 $\mu$l was injected for LC-MS/MS analysis on an UltiMate 3000 RSLCnano system (Thermo Fisher Scientific) in-line connected to a Q Exactive mass spectrometer (Thermo Fisher Scientific). The peptides were first loaded on a trapping column (made in-house, 100 $\mu$m internal diameter [I.D.] × 20 mm, 5-$\mu$m beads, C18 Reprosil-HD, Dr. Maisch) with loading solvent (0.1% TFA in water/ACN, 2/98 [vol/vol]). After 4 min, a valve switch put the loading column in-line with the analytical pump to load the peptides on a 50-cm $\mu$PAC column with C18-endcapped functionality (PharmaFluidics) kept at a constant temperature of 35°C. Peptides were separated with a non-linear gradient from 98% solvent A' (0.1% FA in water) to 30% solvent B' (0.1% FA in water/ACN, 20/80 [vol/vol]) in 70 min, further increasing to 50% solvent B' in 15 min before reaching 99% solvent B' in another 1 min. The column was then washed at 99% solvent B' for 5 min and equilibrated for 15 min with 98% solvent A'. The flow rate was kept constant at 300 nl/min for the entire run, except for the first 9 min during which the flow rate was set to 750 nl/min.

The mass spectrometer was operated in data-dependent, positive ionization mode, automatically switching between MS and MS/MS acquisition for the five most abundant peaks in a given MS spectrum. The source voltage was 2.6 kV, and the capillary temperature was 275°C. One MS1 scan (m/z 400–2,000, AGC target $3 \times 10^6$ ions, maximum ion injection time 80 ms), acquired at a resolution of 70,000 (at 200 m/z), was followed by up to 5 tandem MS scans (resolution 17,500 at 200 m/z, AGC target $5 \times 10^4$ ions, maximum ion injection time 80 ms, isolation window 2 m/z, fixed first mass 140 m/z, spectrum data type: centroid) of the most intense ions fulfilling predefined selection criteria (intensity threshold 1.3 x $10^4$, exclusion of unassigned, 1, 5–8, >8 positively charged precursors, peptide match preferred, exclude isotopes on, dynamic exclusion time 12 s). The HCD collision energy was set to 25% normalized collision energy, and the polydimethylcyclosiloxane background ion at 445.120025 D was used for internal calibration (lock mass).

The generated MS/MS spectra were processed with MaxQuant (version 1.6.1.3) using the Andromeda search engine with default search settings, including a FDR set at 1% on both the peptide and protein levels. Spectra were searched against the sequences of the human proteins in the Swiss-Prot database (release Jan 2019). Matching between runs was enabled and performed on the level of pre-fractionated peptides (fractions 1–10) such that it was enabled for each preceding and each following peptide fraction. The enzyme specificity was set at trypsin/P, allowing for two missed cleavages. Variable modifications were set to oxidation of methionine residues and N-terminal protein acetylation. Carbamidomethylation of cysteine residues was put as a fixed modification. All other settings were kept standard. Proteins were quantified by the

MaxLFQ algorithm integrated into MaxQuant software (74). A minimum of two ratio counts and both unique and razor peptides were considered for protein quantification, leading to the identification of 3,933 proteins across all samples.

Further data analysis was performed with Perseus software (75) (version 1.6.3.4) after uploading the protein group file from MaxQuant. Proteins only identified by site and reverse database hits were removed, as well as potential contaminants. To be able to perform GO, KEGG, Pfam, Corum, and Keyword term enrichment, available annotations were uploaded from the *Homo sapiens* database (release date 2018-04-06). The replicate samples were grouped, and the LFQ intensities were $\log_2$-transformed. Proteins with less than two valid values in at least one group were removed, and missing values were imputed from a normal distribution around the detection limit (with 0.3 spread and 1.8 down-shift). This led to the identification of 3,045 proteins. The efficiency of the cytosol enrichment was evaluated by loading all identified UniProt identifications in the Retrieve/ID Mapping tool on their website and checking the amount of proteins listed with the GOCC term cytosol.

### Selection of N-terminal proteoform pairs for interactome analysis

N-terminal peptides selected using the KNIME workflow outlined in Reference (43) were further annotated and curated in R version 4.1.0.

Part I—annotation of genomic features. Human gene and transcript annotations were downloaded from Ensembl BioMart Archive Release 98 (September 2019), and proteoform accessions were linked to gene identifiers, names, biotypes, and transcript support levels. Subsequently, N-terminal peptides were mapped to genomic positions and a peptide BED file was created. For this purpose, N-terminal peptides matching a Ribo-seq–derived protein accession were represented as transcript coordinates and converted to genomic coordinates using the Proteoformer output SQL database (for TIS distance to transcript start), and R packages biomaRt (2.50.3), GenomicFeatures (1.46.5), rtracklayer (1.54.0), and RMariaDB (1.2.2). Genomic coordinates were generated for UniProt-mapped peptides only if the entire UniProt proteoform was identical to the Ensembl annotated proteoform. Therefore, 109 peptides could not be mapped to the genome. Alternative translation initiation events (aTIS, >1 TIS in the same exon) were distinguished from otherwise possible alternative splicing (TISs in different exons) by extracting the exon ranks of each N-terminal proteoform start site and the corresponding aTIS on the same transcript (if available). Furthermore, we report the exact N-terminal proteoform start codon, frame, and distance to the corresponding aTIS.

Part II—annotation and prediction of protein sequence features. UniProt annotations of human canonical proteins were downloaded in January 2020. Positions of the following features: signal peptide, transit peptide, propeptide, region, motif, coiled coil, compositional bias, repeat, and zinc finger, were converted to protein sequence ranges using GenomicRanges (version 1.46.1). To determine sequence features annotated in UniProt that N-terminal proteoforms had lost, N-terminal peptides were exactly matched to canonical human UniProt proteins using dbtoolkit (version 4.2.5) (76). For N-terminal peptides that mapped internally to a canonical

proteoform, we determined the lost (N-terminal truncated) region. For 5′UTR-extended proteoforms (compared with the canonical proteoform), we considered any sequence feature spanning position 1 or 2 in the canonical protein to be no longer N-terminal in the 5′ extended proteoform and thus lost. Otherwise, for N-terminal peptides not exactly mapping to any UniProt protein, alignment of the entire N-terminal proteoform to one UniProt reference sequence was performed (see Table S3 columns "proteoform.sequence" and "sequence.feature.reference.ID," respectively) using Biostrings, version 2.62.0, pairwiseAlignment function with the following parameters: substitutionMatrix = BLOSUM62, gapOpening = −9.5, gapExtension = −0.5, scoreOnly = FALSE, type = "overlap." The selection criteria of the reference sequence for alignment were as follows: UniProt isoforms were aligned to the matching canonical accession. Ensembl proteoforms were aligned to the UniProt protein from the same gene. In the absence of any reliable UniProt reference (e.g., Ensembl NTR proteoforms), no alignment was performed. From these alignments, sequence ranges that were lost compared with the UniProt reference protein were extracted. We overlapped such lost regions (from exact matching or alignment) with sequence features in the UniProt database (see Table S3 column "lost.UniProtDB.sequence.feature"). We also determined whether N-terminal truncated proteoforms could be derived from N-terminal processing (removed signal, transit, or propeptide) considering a ± one amino acid margin of error (see Table S3 column "signal.transit.propeptide.processing_distance"). For N-terminal peptides without an exact match to a UniProt protein, we scanned for short linear motifs in the proteoform and their selected UniProt reference using ELM resource API (77). From the alignments described above, we extracted sequence stretches that are lost compared with the UniProt reference proteins or sequence stretches gained by the proteoform, and reported the ELMs predicted in these regions (see Table S3 columns "lost_ELM" and "gained_ELM," respectively). TopFIND (78) was used to determine whether the N-terminal peptides could have been derived from post-translational processing (thus leaving out co-translational processing by methionine aminopeptidases; TopFIND results are presented in Table S3 columns "Cleaving.proteases" and "Distance.to.last.transmembrane.domain.shed"). In addition, N-terminal proteoforms that could derive from N-terminal dipeptidase cleavage are marked; see Table S3 column "dipeptidase." Finally, to fully explore sequence similarity of N-terminal proteoforms without an exact UniProt match to other known or predicted proteins in the NCBI database, we performed a BLAST analysis against the human UniProt or the human non-redundant proteins (NCBI, July 2020). Hits with over 80% of protein sequence identity over 50% of the length are reported (see Table S3 columns "blast.versus.uniprot" and "blast.versus.nonredundant"). The molecular weights of N-terminal proteoforms and their annotated counterparts were calculated using the R package Peptides, version 2.4.4.

Part III—survey of complementary data(bases). Curated, experimentally determined protein interactions were downloaded from BioGRID, version 3.5.182 (48), and matched by the gene name, and are reported in Table S3 column "biogrid." Human genetic phenotypes and disorders available from OMIM (79) were matched by gene identifier using biomaRt version 2.50.3, release 98 (see Table S3 column "omim"). Prior experimental evidence for N-terminal proteoform expression in human (primary) cells reported by Van

Damme et al was included for matching N-termini (allowing for Met processing; see Table S3 column "VanDamme.2014") (5). Cytosolic expression associated with the matching gene is reported in Table S3 column "cytosolic.in.HEK.proteome" when confirmed by three independent sources: our own cytosolic proteomics data (see above), gene ontology GOSlim annotation of cellular component (containing "cytosol" and "cytoplasm," while excluding "organelle") (80), and cytosolic subcellular localization determined by immunostaining from the Human Protein Atlas (81).

Genes with multiple proteoforms were classified into four categories: 1. annotated + alternative TIS; 2. multiple alternative TIS; 3. multiple annotated TIS; and 4. one TIS, where canonical UniProt and Ensembl aTIS were considered annotated TIS. Subsequently, we calculated a TIS score for each alternative N-terminal proteoform. High-confident TIS (according to our KNIME workflow), >50% acetylation, spectral count >1, several N-terminal peptides pointing to the same N-terminal proteoform, database or Ribo-seq evidence for TIS, identification in multiple protease conditions (trypsin, chymotrypsin, or endoproteinase GluC), non-AUG start codon in 5'UTR, lost or gained ELM, presence in the Van Damme et al 2014 dataset, genetic disease from OMIM, or no interactions known in BioGRID all increased the score by 1. Truncation of less than 50% of protein length and a UniProt domain lost also increased the score by 3. The score, however, dropped to 0 when we suspected protease cleavage (from TopFIND, through dipeptidase activity, and signal, transit, or propeptide processing) gave rise to the N-terminal proteoform. A gene score was further calculated as the sum of N-terminal proteoform scores of a given gene. Gene scores were kept only for genes of the following categories: 1. annotated + alternative TIS; 2. multiple alternative TIS; and 4. one TIS, if we had orthogonal evidence of cytosolic localization, thus prioritizing genes with novel proteoforms. If the gene score was >1, it is included in Table S3 (Tab 2, gene score >0).

The list of 372 genes with a score >1 was trimmed to a manageable list for Virotrap analysis using prioritization and selection criteria. These criteria, ranked according to their importance, are the following: (1) high-confident N-terminal proteoforms are prioritized over low-confident proteoforms. Either the former are assigned as high confident following our KNIME filtering strategy or for which there was extra evidence supporting their synthesis. Such evidence can be either a known UniProt isoform, extra Ribo-seq evidence supporting the TIS, its reporting in the dataset of Van Damme et al (5), or its detection on WBs as available in the Human Protein Atlas (see the next paragraph for details). (2) Proteoforms with a loss or gain of a known domain or ELM were given higher priority. (3) Proteoforms with truncation or extension of >20 amino acids or >50 amino acids for long proteins (>700 amino acids) were preferred over shorter truncations. (4) The more the evidence for the cytosolic localization of the proteoform, the higher its priority as Virotrap is restricted to cytosolic proteins. To increase the overall confidence for the cytosolic localization of a proteoform, we also relied on our own cytosolic proteome map. (5) Non-structural proteins such as enzymes were prioritized over structural proteins (e.g., cytoskeletal proteins) as Virotrap encountered difficulties using structural proteins as baits. (6) Proteins that have a known association with a human disease according to the OMIM database were prioritized over proteins without such a link. (7)

Proteoforms resulting from translation out-of-frame from the start site of the canonical protein sequence were considered as novel proteins, thus not as N-terminal proteoforms, and were not selected for further analysis.

To further evaluate the identified N-terminal proteoforms, we used WB data available in the Human Protein Atlas (https://www.proteinatlas.org/). From the 372 genes with a score >1, we evaluated whether the N-terminal proteoform could be detectable on WB (based on the actual mass difference and the likelihood that the epitope recognized by the antibody is still present in the proteoform). For 138 of the 372 genes, the N-terminal proteoforms would potentially be detectable and these genes were further evaluated here. Information on all antibodies tested for the protein products of these genes (also including unpublished antibodies) was retrieved from the Human Protein Atlas LIMS. A WB score was retrieved. This score ranges between 1 and 7 and points to the following: 1, single band corresponding to the predicted size (±20%); 2, band of predicted size (±20%) with additional bands present; 3, single band larger than predicted size (+20%), but partly supported by experimental and/or bioinformatics data; 4, no bands detected; 5, single band differing more than ±20% from predicted size and not supported by experimental and/or bioinformatics data; 6, weak band of predicted size but with additional bands of higher intensity also present; and 7, only bands that do not correspond to the predicted size (extra information can be found here: https://www.proteinatlas.org/learn/method/western+blot). Genes yielding protein products with a WB score of 2, for at least one of the antibodies raised again these protein products, were further checked. Although scores 6 and 7 also indicate extra bands (and thus potential N-terminal proteoforms), the confidence in these results is lower and these were thus not considered further. For the genes giving rise to protein products with a score of 2, the WBs generated were checked on the Human Protein Atlas website (www.proteinatlas.org) for a protein band with a size corresponding to the N-terminal proteoform identified. When such bands were found, this further increased the overall confidence score for this N-terminal proteoform and was thus considered in the final selection.

The prioritization/selection and the Human Protein Atlas information reduced the list of genes to 85 (pairs of canonical proteins and one or two proteoforms), and from this list, 22 genes were selected for Virotrap analysis that were all top-ranked as they met most to all selection criteria and some also contained the Human Protein Atlas WB evidence.

## Tissue expression of N-terminal proteoforms evaluated through re-analysis of public proteomics data

Mass spectrometry data of the draft human proteome map developed by the Pandey group (40), composed of 30 histologically normal human samples including 17 adult tissues, 7 fetal tissues, and 6 purified primary hematopoietic cells, were downloaded from PRIDE project PXD000561 and searched with ionbot, version 0.8.0 (41 Preprint). Of the 30 samples, each was processed by several sample preparation methods and MS acquisition pipelines to generate 84 technical replicates. We first generated target and decoy databases from our custom-built database containing UniProt canonical and

isoform entries appended with Ribo-seq–derived protein sequences (36). Next, we searched the mass spectrometry data with semi-tryptic specificity, DeepLC retention time predictions (82) and protein inference enabled, precursor mass tolerance set to 10 ppm, and a q-value filter of 0.01. Carbamidomethylation of cysteines was set as a fixed modification, oxidation of methionines and N-terminal acetylation were set as variable modifications, and an open-modification search was disabled. Downstream analysis was performed in R, version 4.1.0, using dplyr (1.0.9), Biostrings (2.26.0), GenomicRanges (1.46.1), and biomaRt (2.50.3, release 98). To constrict the results to the first-ranked PSM per spectrum, we used ion-bot.first.csv output and filtered out decoy hits, common contaminants that do not overlap with target FASTA and used PSM q-value ≤ 0.01. Because of the complexity of our custom protein database, most PSMs were associated with several protein accessions. We sorted accessions to prioritize UniProt canonical followed by UniProt isoforms, followed by Ribo-seq, higher peptide count (in the whole sample), start (smallest start position first), and accession (alphabetically). These steps yielded a filtered PSM table. Subsequently, we sorted PSMs by N-terminal modification (to prioritize N-terminally acetylated peptidoforms) and the highest PSM score. Sorted PSMs were grouped by matched peptide sequence yielding a unique peptide table. Peptides were grouped by sorted accession to generate a protein table, complemented with sample and protein metadata (such as gene and protein names, descriptions). Per sample and replicate, we obtained a unique peptide count, spectral count, and NSAF quantification. Briefly, SAF values were first calculated per protein based on total spectral counts corrected for protein length (because longer proteins produce more peptides). Next, to accurately account for variation between samples, individual SAF values were divided by the sum of all SAFs per sample, resulting in the NSAF value. Log transformation of NSAF values was performed to achieve normal distribution of data suitable for downstream statistical analysis. Differential expression analysis across all tissues was performed using limma (3.50.3) based on $\log_2$ NSAF values only for proteoforms found in all replicates of at least one tissue (9,644/26,159 proteoforms). Importantly, for non-canonical proteoforms, only peptides that do not map to canonical UniProt proteins via dbtoolkit (version 4.2.5) (76) were considered for the quantification. We extracted pairwise contrasts between adult versus adult, fetal versus fetal, and adult versus fetal of the same tissue, considering only significant differences in expression with a Benjamini–Hochberg adjusted *P*-value of 0.05. Boxplots were created using ggplot2 (3.3.6), whereas heatmaps of N-terminal proteoform expression were generated using pheatmap (1.0.12). To determine the row clustering, we used $\log_2$ NSAF values converted to binary data as input for MONothetic Analysis (cluster version 2.1.3).

### Generation of Virotrap clones

GAG–bait fusion constructs were generated as described (35). The coding sequences for the full-length protein were either ordered from IDT (gBlocks gene fragments, as was the case for the following constructs: *CAPRIN1*, *SPAST*, *PRUNE*, *SORBS3*, *FNTA*, *CAST*, *RARS*, *UBXN6*, *PAIP1*, *PXN*, and NTR protein) or generated cDNA was used as a template for PCR amplification using AccuPrime *pfx* DNA polymerase (Invitrogen) and ORF-specific primers. cDNA was generated

by isolating RNA from $5 \times 10^6$ HEK293T cells with the NucleoSpin RNA isolation mini kit (Macherey-Nagel) according to the manufacturer's instructions. 500 ng of isolated RNA was then used as input for generating cDNA, and cDNA synthesis was performed using the PrimeScript RT kit (Takara Bio) according to the manufacturer's instructions. The generated PCR products for the full-length proteins were transferred into the pMET7-GAG-sp1-RAS plasmid by classic cloning with restriction enzymes (EcoRI and XbaI) or In-Fusion seamless cloning (Takara Bio) when the genes contained internal restriction sites for EcoRI and XbaI (which was the case for *CSDE1*, *NTR*, and *UBAC1*).

The N-terminal proteoforms were amplified from the corresponding generated pMET7-GAG-sp1-FL plasmid of each gene using the AccuPrime *pfx* DNA polymerase (Invitrogen) with proteoform-specific primers. However, some proteoforms also contain extensions or large internal deletions and these were ordered from IDT (gBlocks gene fragments, as was the case for the following constructs: the two N-terminal proteoforms of CAST [P20810-4 and P20810-8], N-terminal proteoform of PXN [P49023-4], an N-terminally extended proteoform of UBE2M, and one of the two N-terminal proteoforms of SPAST [Q9UBP0-4]). The generated PCR products of the proteoforms were transferred into the pMET7-GAG-sp1-RAS plasmid as indicated above.

### Virotrap studies

For full details on the Virotrap protocol, we refer to Reference (35). HEK293T cells were kept at low passage (<10) and cultured at 37°C and 8% $CO_2$ in DMEM, supplemented with 10% fetal bovine serum, 25 U/ml penicillin, and 25 $\mu$g/ml streptomycin.

For WB validation of expression, $1.15 \times 10^6$ HEK293T cells were seeded in six-well plates the day before transfection. On the day of transfection, a DNA mixture was prepared containing the following: 0.82 $\mu$g bait construct (pMET7–GAG–sp1–bait), 0.046 $\mu$g pMD2.G, and 0.093 $\mu$g pcDNA3-FLAG-VSV-G. In each experiment, eGFP and eDHFR were taken along as controls. For eGFP, normal expression amounts were used instead of maximal expression as used for the baits. For eDHFR, two expression amounts were used to allow a comparison of bait intensities with control intensities. For normal expression, a DNA mixture was prepared containing the following: 0.48 $\mu$g control construct (either pMET7-GAG-sp1-eGFP or pMET7-GAG-sp1-eDHFR), 0.34 $\mu$g of pSVsport (mock vector), 0.046 $\mu$g pMD2.G, and 0.046 $\mu$g pcDNA3-FLAG-VSV-G. Cells were transfected using polyethyleneimine (PEI). After 6 h of transfection, the medium was refreshed with 2 ml of fresh growth medium. After 46 h, the cellular supernatant was collected and the cellular debris was removed from the harvested supernatant by 3-min centrifugation at 400*g* at room temperature. The cleared medium was then incubated with 10 $\mu$l Dynabeads MyOne Streptavidin T1 beads (Invitrogen) pre-loaded with 1 $\mu$g monoclonal Anti-FLAG BioM2-Biotin, Clone M2 (Sigma-Aldrich), according to the manufacturer's protocol. After 2-h binding at 4°C by end-over-end rotation, beads were washed twice with washing buffer (20 mM Hepes, pH 7.5, and 150 mM NaCl) and the captured particles were released directly in 40 $\mu$l SDS–PAGE loading buffer. A 10-min incubation step at 65°C before removal of the beads (by binding them to the magnet) ensured complete release and lysis of the VLPs.

Lysates of the producer cells were prepared by scraping the cells in 100 $\mu$l Gingras lysis buffer (50 mM Hepes–KOH, pH 8.0, 100 mM KCl, 2 mM EDTA, 0.1% NP-40, and 10% glycerol supplemented with 1 mM DTT, 0.5 mM PMSF, 50 mM glycerophosphate, 10 mM NaF, 0.25 mM sodium orthovanadate, and cOmplete protease inhibitor cocktail [Roche]) after washing of the cells in chilled PBS. The lysates were cleared by centrifugation at 13,000$g$, 4°C for 15 min, to remove the insoluble fraction. The protein concentration was measured, and 25 $\mu$g of protein material was mixed with SDS–PAGE loading buffer (diluted to a total volume of 30 $\mu$l). After heating to 95°C for 5 min, both supernatant and lysate samples were loaded on Criterion XT 4–12% Bis-Tris gels (Bio-Rad Laboratories). Each set of experiments also contained the GAG-EGFP expression control. After SDS–PAGE, proteins were transferred to a PVDF membrane (Merck Millipore) after which the membrane was blocked using Odyssey blocking buffer (LI-COR) diluted once with TBS-T (TBS supplemented with 0.1% Tween-20). Immunoblots were incubated overnight with primary antibodies against GAG (Abcam) and, for the lysates, also with primary antibody against actin (Sigma-Aldrich), serving as a loading control, in Odyssey blocking buffer (PBS) diluted once with TBS-T. Blots were washed four times with TBS-T and incubated with fluorescently labeled secondary antibodies, IRDye 800CW Goat Anti-Mouse IgG polyclonal 0.5 mg (LI-COR) and IRDye 680RD Goat Anti-Rabbit IgG polyclonal 0.5 mg (LI-COR) in Odyssey blocking buffer diluted once with TBS-T for 1 h. After three washes with TBS-T and an additional wash in TBS, immunoblots were imaged using the Odyssey infrared imaging system (LI-COR).

For LC-MS/MS analysis, all baits were divided into sets of maximally four baits and one control bait based on similar expression levels as judged from WB data. In total, seven sets of baits were generated, and per set, always a FL and a PR bait were included unless specified otherwise. The first set included eDHFR, CSDE1, MAVS, and TSC22D3; the second set, eDHFR, PAIP1, UBXN6, and ZFAND1; the third set, eDHFR, SPAST (FL, PR1, and PR2), and UCHL1 (FL, PR1, and PR2); the fourth set, eDHFR, CFL2, and FNTA; the fifth set, eDHFR, AIMP1, PRPSAP1, and UBE2M; the sixth set, eDHFR, PRUNE1, PXN, UBAC1, and a protein from a non-translated region; and the last set, eDHFR, CACYBP, CAPRIN1, and EIF4A1. Each construct was analyzed in triplicate, and for every replicate, the day before transfection, a 75-cm$^2$ falcon was seeded with 9 × 10$^6$ cells. Cells were transfected using PEI, with a DNA mixture containing 6.43 $\mu$g of bait plasmid (pMET7–GAG–bait), 0.71 $\mu$g of pcDNA3-FLAG-VSV-G plasmid, and 0.36 $\mu$g of pMD2.G plasmid. Based on the WB results, we decided to use normal expression levels of eDHFR as their intensities were most similar to the intensities of the different baits. Thus, for the eDHFR control, cells were transfected with a DNA mixture containing 3.75 $\mu$g of eDHFR plasmid (pMET7-GAG-eDHFR), 2.68 $\mu$g of pSVsport plasmid, 0.71 $\mu$g of pcDNA3-FLAG-VSV-G plasmid, and 0.36 $\mu$g of pMD2.G plasmid. The medium was refreshed after 6 h with 8 ml of DMEM supplemented with 10% FBS, 25 U/ml penicillin, and 25 $\mu$g/ml streptomycin.

The cellular supernatant (containing the VLPs that engulf the bait and its associated interaction partners) was harvested after 46 h and centrifuged for 3 min at 1,250$g$ to remove debris. The cleared supernatant was then filtered using 0.45-$\mu$m filters (Merck Millipore). For every sample, 20 $\mu$l MyOne Streptavidin T1 beads in suspension (10 mg/ml; Thermo Fisher Scientific) were first washed with 300 $\mu$l wash buffer containing 20 mM Tris–HCl, pH 7.5, and 150 mM NaCl, and subsequently pre-loaded with 2 $\mu$l biotinylated anti-FLAG antibody (BioM2; Sigma-Aldrich). This was done in 500 $\mu$l wash buffer, and the mixture was incubated for 10 min at room temperature. Beads were added to the samples, and the VLPs were allowed to bind for 2 h at room temperature by end-over-end rotation. Bead–particle complexes were washed once with 200 $\mu$l washing buffer (20 mM Tris–HCl, pH 7.5, and 150 mM NaCl) and subsequently eluted with a FLAG peptide (30 min at 37°C; 200 $\mu$g/ml in washing buffer). VLPs were lysed by addition of Amphipol A8–35 (Anatrace) (83) to a final concentration of 1 mg/ml. After 10 min, the lysates were acidified (pH < 3) by adding 2.5% FA. Samples were centrifuged for 10 min at >20,000$g$ to pellet the protein/Amphipol A8–35 complexes. The supernatant was removed, and the pellet (containing the proteins) was re-suspended in 20 $\mu$l 50 mM fresh triethylammonium bicarbonate (TEAB). Proteins were heated at 95°C for 5 min, cooled on ice to room temperature for 5 min, and digested into peptides overnight at 37°C with 0.5 $\mu$g of sequencing-grade trypsin (Promega). Peptide mixtures were acidified to pH 3 with 1.5 $\mu$l 5% FA. Samples were centrifuged for 10 min at 20,000$g$, and 7.5 $\mu$l of the sample was injected for LC-MS/MS analysis on an UltiMate 3000 RSLCnano system in-line connected to a Q Exactive HF Biopharma mass spectrometer (Thermo Fisher Scientific).

For each sample, 7.5 $\mu$l of the supernatant was injected for LC-MS/MS analysis on an UltiMate 3000 RSLCnano system in-line connected to a Q Exactive HF Biopharma mass spectrometer (Thermo Fisher Scientific). Trapping was performed at 10 $\mu$l/min for 4 min in loading solvent A on a 20-mm trapping column (made in-house, 100 $\mu$m internal diameter [I.D.], 5-$\mu$m beads, C18 Reprosil-HD, Dr. Maisch, Germany). The peptides were separated on a 250-mm Waters nanoEase M/Z HSS T3 Column, 100 Å, 1.8 $\mu$m, 75 $\mu$m inner diameter (Waters Corporation), kept at a constant temperature of 50°C. Peptides were eluted by a non-linear gradient starting at 1% MS solvent B reaching 55% MS solvent B in 80 min and 97% MS solvent B in 90 min, followed by a 5-min wash at 97% MS solvent B and re-equilibration with MS solvent A. The mass spectrometer was operated in a data-dependent mode, automatically switching between MS and MS/MS acquisition for the 12 most abundant ion peaks per MS spectrum. Full-scan MS spectra (375–1,500 m/z) were acquired at a resolution of 60,000 in the Orbitrap analyzer after accumulation to a target value of 3,000,000. The 12 most intense ions above a threshold value of 13,000 were isolated with a width of 1.5 m/z for fragmentation at a normalized collision energy of 30% after filling the trap at a target value of 100,000 for a maximum 80 ms. MS/MS spectra (200–2,000 m/z) were acquired at a resolution of 15,000 in the Orbitrap analyzer.

The generated MS/MS spectra were processed with MaxQuant (version 1.6.17.0) using the Andromeda search engine with default search settings, including a FDR set at 1% on both the peptide and protein levels. For the Virotrap experiment, the sequences of the human proteins in the Swiss-Prot database (release Jan 2021) were complemented with the sequences of GAG, VSV-G, FLAG-VSVG, and eDHFR and the sequences of the baits were replaced by the sequences of their shortest proteoforms (only the baits used in that set were adjusted) and used as a search space. The replacement of the bait sequences to the shortest variant was done to allow more

correct quantification of these proteins over the full-length and proteoform bait interactomes.

The enzyme specificity was set at trypsin/P, allowing for two missed cleavages. Variable modifications were set to oxidation of methionine residues and N-terminal protein acetylation. Other settings were kept as standard unless specified here. Proteins were quantified by the MaxLFQ algorithm integrated into MaxQuant software. MaxLFQ values are normalized between samples based on a population of proteins that are presumed to change minimally between experimental conditions (74). VLPs contain numerous background proteins that are commonly identified and stable providing a good background for MaxLFQ normalization. Peptides with a minimum of two ratio counts of both unique and razor peptides were considered for protein quantification. Further data analysis was performed with Perseus software (75) (version 1.6.15.0) after uploading the protein group file from MaxQuant. Proteins only identified by site and reverse database hits were removed, as well as potential contaminants. Replicate samples were grouped, and proteins with less than three valid values in at least one group were removed. Missing values were imputed using imputeLCMD (R package implemented in Perseus) with a truncated distribution with parameters estimated using quantile regression (QRILC).

The matrix containing all identified peptides (filtered on valid values and with imputed LFQ values) was exported from Perseus, and statistical analysis was performed on Genstat (version V21; https://genstat21.kb.vsni.co.uk/). For each set, proteins were analyzed separately by fitting a linear model of the following form: response = $\mu$ + bait + error, where the response represents the $\log_2$-transformed LFQ intensity measured. The significance of the bait effect was assessed by a F test, and the significance of individual comparisons between the bait factor was assessed by a t test. The performed pairwise comparisons are tabulated in Table S11. Correction for multiple testing was done by estimating the FDR by modeling significance values as a 2-component mixture of uniform and B or $\Gamma$ densities as implemented in Genstat v21.

Potential interaction partners of all baits (both full-length proteins and N-terminal proteoforms) were selected in pairwise contrasts between the bait and eDHFR control samples at a FDR of 0.01. The generated list of potential interaction partners was compared with known interaction partners listed in BioGRID (48), STRING (84), and IntAct (49). Pairwise contrasts of interest between proteoforms were selected at a FDR of 0.05 and a difference of at least one $\log_2$ change. Furthermore, when the difference in prey levels between the proteoform interactomes was at least twofold, the difference required in order to be retained needed to be higher than the difference in levels between the bait proteoforms (on the side of the most intense proteoform). Only proteins also listed as candidate interaction partners in the comparisons of eDHFR control samples with the baits (FL and PR) were retained as potential differential interaction partners of the proteoforms.

## Generation of Y2H clones

The generation of Y2H clones was done as described (8). All 22 selected pairs of full-length proteins and proteoforms were cloned into four different Y2H expression vectors by Gateway Cloning (Invitrogen). The full-length proteins and their N-terminal

proteoforms were amplified from the corresponding pMET7-GAG-sp1-FL or pMET7-GAG-sp1-PR plasmids of each gene using the AccuPrimeTM pfx DNA polymerase (Invitrogen) and ORF-specific primers supplemented with attB sites. PCR products were transferred into pDONR221 by Gateway BP reaction (using BP Clonase II Enzyme Mix, Invitrogen) according to the manufacturer's instructions to generate entry clones. Entry clones were transformed and sequenced according to the manufacturer's instructions. Then, all proteins were transferred from entry clones into each of the four destination vectors (pDEST-AD, pDEST-AD-AR68, pDEST-AD-QZ213l, and pDEST-DB; see Reference (44) for vector details) by Gateway LR reaction (Gateway LR Clonase II Enzyme Mix; Invitrogen) according to the manufacturer's instructions. The resulting expression vectors were used for Y2H.

## Y2H experiments

Y2H screening was performed as described (44). All baits (coupled to the Gal4 DNA-binding domain, DB) were tested against the hORFeome v9.1 collection of ~17,408 ORF clones fused to the Gal4 activation domain (AD). After first-pass screening, each bait was pairwise tested for interaction with the identified candidate partners. In this step, the interacting partners of any proteoforms of a gene were tested against all the proteoforms of that gene to eliminate false negatives because of sampling sensitivity. Pairs showing a positive result in the pairwise retest were PCR-amplified and sequence-confirmed (Sanger, Azenta) to confirm the identity of clones encoding each interacting protein.

## Generation of clones for AP-MS

For EIF4A1, MAVS, and PAIP1, both the canonical proteins and N-terminal proteoforms of pMET7–bait–tag proteins were constructed via an in-house–developed Golden Gate assembly platform.

The following table gives an overview of all generated constructs: pMET7_MAVS_FL_FLAG, pMET7_MAVS_PR_FLAG, pMET7_PAIP1_FL_FLAG, pMET7_PAIP1_PR_FLAG, pMET7_EIF4A1_FL_FLAG, pMET7_EIF4A1_PR_FLAG, and pMET7_eDHFR-FLAG.

## AP-MS

For AP-MS experiments, per experiment, $1.5 \times 10^7$ HEK293T cells were seeded the day before transfection in a 150-mm dish. 7.5 $\mu$g of bait–FLAG DNA or 7.5 $\mu$g of eDHFR-FLAG DNA was transfected using PEI as described above. 40 h post-transfection, cells were washed with ice-cold PBS and scraped in 1.5 ml of lysis buffer (containing 10 mM Tris–HCl, pH 8, 150 mM NaCl, 1% NP-40, 10% glycerol, 1 mM sodium orthovanadate, 20 mM $\beta$-glycerophosphate, 1 mM NaF, 1 mM PMSF, and cOmplete protease inhibitor cocktail). Cells were lysed on ice for 30 min and subsequently centrifuged for 15 min at 4°C at 20,000g. The supernatant was transferred to a new 1.5-ml protein LoBind Eppendorf tube, and the protein concentration was measured using a Bradford assay. Sample volumes were adjusted with lysis buffer so that all samples were at the same protein concentration.

For every experiment, 10 µl MyOne Protein G beads (Invitrogen) were pre-loaded with 1 µg anti-FLAG (Sigma-Aldrich) and added to 350 µl of the cleared supernatant. Protein complexes were allowed to bind for 2 h by end-over-end rotation at 4°C. The beads were washed twice with lysis buffer and three times with 20 mM Tris–HCl, pH 8.0, and 2 mM CaCl$_2$. Beads were re-suspended in 25 µl 20 mM Tris–HCl, pH 8.0, and overnight incubated with 1 µg sequencing-grade modified trypsin at 37°C. After removal of the beads, the samples were incubated for another 3 h with 250 ng trypsin. Samples were then acidified by 2% FA (f.c.) before LC-MS/MS analysis. Peptides were analyzed by LC-MS/MS using a Thermo Fisher Scientific Q Exactive HF operated similarly as described above for Virotrap. The generated MS/MS spectra were processed similarly as described for Virotrap (see above). Different here is that a newer version of MaxQuant was used (2.1.4.0) and that the database was supplemented with the sequence of eDHFR-FLAG. Overall, 2,705 proteins were identified over all samples. Based on the LC-MS/MS profiles, three samples were not further considered for analysis (eDHFR replicate C, PAIP1 FL replicate C, and EIF4A1 FL replicate C).

Further data analysis was performed with Perseus software ([75]) (version 1.6.15.0) after uploading the protein group file from Max-Quant. Proteins only identified by site and reverse database hits were removed, as well as potential contaminants. Replicate samples were grouped, and proteins with less than three valid values in at least one group were removed, reducing the matrix to 1,903 protein identifications. Missing values were imputed using impu-teLCMD (R package implemented in Perseus) with a truncated distribution with parameters estimated using quantile regression (QRILC). Imputed log$_2$ LFQ values of AP-MS data were subjected to statistical analysis in R using limma, version 3.50.3. Pairwise contrasts of interest between the different bait samples were retrieved at a significance level of $\alpha$ 0.05, corresponding to a Benjamini–Hochberg adjusted $P$-value (FDR) cutoff.

## Data Availability

The mass spectrometry proteomics data have been deposited to the ProteomeXchange Consortium via the PRIDE ([85]) partner repository with the following dataset identifiers:

- PXD030601 (cytosolic N-terminal COFRADIC data from the three different proteases, searched with the custom database)
- PXD039392 (N-terminal COFRADIC on cytosolic proteins of HEK293T cells—UniProt search)
- PXD039127 (mapping the cytosolic proteins of HEK293T cells)
- PXD039171 (Virotrap-based interactome analysis of 20 pairs of N-terminal proteoform(s) and their canonical counterpart)
- PXD039085 (AP-MS data)
- PXD039339 (Tissue expression of N-terminal proteoforms, re-analysis "Draft map of the human proteome").

The analysis code was made available in the following GitHub repositories: https://github.com/dfjlkw/Nt-proteoform-R-workflow.git and https://github.com/dfjlkw/Tissue-expression-Nt-proteoforms.git.

## Supplementary Information

## Acknowledgements

This work was supported by the Research Foundation—Flanders (FWO), project numbers G008018N and G002721N (to K Gevaert). The authors would like to thank Katie Boucher, Evy Timmerman, and Francis Impens from the VIB Proteomics Core (VIB-UGent) for operating the MS instrument.

### Author Contributions

A Bogaert: conceptualization, formal analysis, validation, investigation, visualization, methodology, and writing—original draft.
D Fijalkowska: conceptualization, formal analysis, validation, investigation, visualization, methodology, and writing—original draft, review, and editing.
A Staes: conceptualization, formal analysis, supervision, investigation, methodology, and writing—review and editing.
T Van de Steene: formal analysis, investigation, methodology, and writing—review and editing.
M Vuylsteke: conceptualization, data curation, formal analysis, methodology, and writing—review and editing.
C Stadler: formal analysis, investigation, and writing—review and editing.
S Eyckerman: conceptualization, investigation, methodology, and writing—review and editing.
K Spirohn: conceptualization, formal analysis, validation, investigation, methodology, and writing—review and editing.
T Hao: formal analysis, validation, investigation, methodology, and writing—review and editing.
MA Calderwood: conceptualization, methodology, and writing—review and editing.
K Gevaert: conceptualization, data curation, supervision, funding acquisition, methodology, project administration, and writing—review and editing.

### Conflict of Interest Statement

The authors declare that they have no conflict of interest.

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
