## [Reviewer comments · Life Science Alliance]

Life Science Alliance

N-terminal proteoforms may engage in different protein complexes

Annelies Bogaert, Daria Fijalkowska, An Staes, Tessa Van de Steene, Marnik Vuylsteke, Charlotte Stadler, Sven Eyckerman, Kerstin Spirohn, Tong Hao, Michael Calderwood, and Kris Gevaert

DOI: <https://doi.org/10.26508/lsa.202301972>

Corresponding author(s): *Kris Gevaert, VIB-UGent Center for Medical Biotechnology*

Review Timeline:	Submission Date:	2023-02-06
	Editorial Decision:	2023-03-24
	Revision Received:	2023-05-24
	Editorial Decision:	2023-05-25
	Revision Received:	2023-05-26
	Accepted:	2023-05-30

Transaction Report:

March 24, 2023

Re: Life Science Alliance manuscript #LSA-2023-01972-T

Kris Gevaert
VIB-UGent
MBC
Ghent 9000

Dear Dr. Gevaert,

Thank you for submitting your manuscript entitled "N-terminal proteoforms may engage in different protein complexes" to Life Science Alliance. The manuscript was assessed by an expert reviewer, whose comments are appended to this letter. We invite you to submit a revised manuscript addressing the Reviewer comments.

When submitting the revision, please include a letter addressing the reviewer comments point by point.

Thank you for this interesting contribution to Life Science Alliance. We are looking forward to receiving your revised manuscript.

Sincerely,

B. MANUSCRIPT ORGANIZATION AND FORMATTING:

Reviewer #1 (Comments to the Authors (Required)):

Summary:

The authors explore N-terminal proteoforms in a mammalian cell line and investigate their impact on protein complex formation. They construct an N-terminal proteoform catalogue of the HEK293T cellular cytosol by mapping peptides generated by COFRADIC to the UniProt database and a custom-built database including Ribo-seq data and UniProt isoforms. They apply a custom scoring system to select a subset of 22 high-confidence N-terminal proteoforms for interaction mapping. The authors show that N-terminal proteoforms have significant tissue-dependent expression profiles in humans and compare the potential interaction partners for candidate N-terminal proteoforms to their canonical counterparts using Virotrap technology.

General remarks:

The authors present a comprehensive study about cytoplasmic, N-terminal proteoforms in the HEK cell cytoplasm. Their approach is convincing and validates the existence of 20 N-terminal proteoforms with divergent PPI interaction partner profiling, which is a valuable addition to the literature. However, the authors use several assumptions along the way from the proteomic measurement to the final list of 20 proteins, which limits the ability to refer back from the 20 proteins to the global nature of the N-proteoformome. I recommend publishing of the article, it's a great contribution to LSA, but including a 'study limitation' section to openly discuss the bias that emerged from the strong multistep selection criteria that reduced a long list of MS hits to just 20 proteins.

Major points:

The authors could include a 'study limitation' section to openly discuss the bias that emerged from the strong multistep selection criteria that reduced a long list of MS hits to just 20 proteins.

The authors could comment or explain the discrepancy in the number of N-terminal proteoforms identified in their study compared to the literature estimates.

The authors could be careful when blaming the low agreement between Y2H and protein capture techniques on Y2H, since Y2H is the *in vivo* technique.

Minor points

Minor and technical comments (including suggestions to the Authors, so that they could improve the presentation and accessibility of their manuscript):

The authors could use consistent language and abbreviation throughout the paper, especially for N-terminal / Nt and N-terminal proteoform / Nt-proteoform.

Abbreviations could be introduced in the Results section as well as in the Materials and Methods section.

The resolution of figures is low, and the color schemes could be friendlier, especially in Figure 6.

Supplementary tables could be named the same as in the manuscript.

Abstract:

The abstract is well written and provides a good overview of the study aim and what was done. However, the authors could elaborate on how Nt-proteoforms originate from splice variants in the introduction section.

Introduction:

The topic is introduced well, and the reader understands the aim and scope of the study, as well as why the topic is of interest. However, some aspects to understand the workflow are not covered well. COFRADIC and Virotrap are mentioned and linked to literature, but they are key parts of the workflow and could be explained in a few sentences. The first three paragraphs are also repetitive in language and need rewording. Additionally, the statement "Nt-proteoforms have long been overlooked and studies on their biological function are emerging just now" needs revision since the cited references are more than 20 years old.

Materials and Methods:

The analysis code should be made available, and the generation of a cytosolic proteome map of HEK293T cells needs better explanation. The authors could clarify how protein/peptide MS quantities were normalized across different tissues when evaluating tissue expression of Nt-proteoforms through re-analysis of public proteomics data. The authors could also introduce

abbreviations in the Materials and Methods section, as some were only introduced in the Results section. For Virotrap studies, the authors could introduce the abbreviations PR and FL at their first use, clarify what was measured, and explain when imputation was applied and why. The authors could also clarify if peptide quantities were normalized between baits and control and why missing potential interaction partners of all baits were imputed.

Results:

In the Interaction profiling of Nt-proteoforms and their canonical counterparts section, the authors could address MaxQuant's normalization strategies and how similar VLP contents were within VLPs. They could also clarify if VLPs of different baits can be analyzed together regarding normalization and imputation. For the Proteoform-specific interaction partners section, the authors could explain the reason for imputation and whether it would be necessary for a more suitable analysis strategy. In the Studying selected differences between proteoform interactomes by AP-MS section, the authors could clarify why some proteins did not show up in the table of candidate interactors for AP-MS but were present in the Volcano plot. They could also evaluate the comparison of identified interaction networks to known interaction networks as mentioned in the text. Lastly, the last paragraphs of the Interaction profiling of Nt-proteoforms and their canonical counterparts section are not well-formulated.

Overall, the authors have done a lot of work and presented a valuable addition to the literature. While there are some issues in the details, they are solvable, and I recommend the publication of this work in LSA.

Reviewer #1 (Comments to the Authors (Required)):

Summary:

The authors explore N-terminal proteoforms in a mammalian cell line and investigate their impact on protein complex formation. They construct an N-terminal proteoform catalogue of the HEK293T cellular cytosol by mapping peptides generated by COFRADIC to the UniProt database and a custom-built database including Ribo-seq data and UniProt isoforms. They apply a custom scoring system to select a subset of 22 high-confidence N-terminal proteoforms for interaction mapping. The authors show that N-terminal proteoforms have significant tissue-dependent expression profiles in humans and compare the potential interaction partners for candidate N-terminal proteoforms to their canonical counterparts using Virotrap technology.

General remarks:

The authors present a comprehensive study about cytoplasmic, N-terminal proteoforms in the HEK cell cytoplasm. Their approach is convincing and validates the existence of 20 N-terminal proteoforms with divergent PPI interaction partner profiling, which is a valuable addition to the literature. However, the authors use several assumptions along the way from the proteomic measurement to the final list of 20 proteins, which limits the ability to refer back from the 20 proteins to the global nature of the N-proteoformome. I recommend publishing of the article, it's a great contribution to LSA, but including a 'study limitation' section to openly discuss the bias that emerged from the strong multistep selection criteria that reduced a long list of MS hits to just 20 proteins.

We would like to thank the reviewer for her/his kind appreciation of our manuscript and support for publishing it. Including a 'study limitation' section is indeed an interesting addition to our manuscript as it provides a more complete view on all aspects of our study. We addressed this comment by adding a part in the discussion section that emphasizes that our selection strategy also induces a bias. The adjustments made are outlined in more detail below.

Major points:

The authors could include a 'study limitation' section to openly discuss the bias that emerged from the strong multistep selection criteria that reduced a long list of MS hits to just 20 proteins.

Please note that in the discussion section, we had already discussed some aspects of our strategy such as the focus on cytosolic proteins and the use of HEK293T cells. We now extended the discussion by explaining possible biases in our selection strategy and by including also other discussion points different from those suggested by the reviewer. Below is an overview of the changes made.

The following part was added:

"We here attempted a more global analysis of the interaction profiles of N-terminal proteoforms and their canonical counterparts. However, as Virotrap, and other MS-based PPI methods, are labor-intensive, only a limited, yet well-selected set of pairs could be analyzed. Funneling all identified N-terminal proteoforms to a manageable set for Virotrap-based interactome analysis was based on several selection criteria that could have introduced biases. First, as mentioned, we focused on cytosolic proteoforms. Second, the identified peptides were stringently filtered to only select N-terminal peptides confidently originating from translation events, thus ignoring N-

terminal proteoforms originating from protein processing, which further reduced the N-terminal proteoforms that could be considered for further analysis. The fraction of N-terminal proteoforms among all identified peptides is much smaller than expected from the composition our database (see figure 3A in reference <https://pubmed.ncbi.nlm.nih.gov/35788065/>). From 22,003 UniProt isoforms, we identified 89, of which 83 were highly confident identifications. In analogy, from 60,661 proteoforms predicted from Ribo-seq data, we could only identify 80, including 73 with high confidence. Similar disproportions have been reported by (N-terminal) riboproteogenomics studies (<https://pubmed.ncbi.nlm.nih.gov/24623590/>, <https://pubmed.ncbi.nlm.nih.gov/25156699/>, <https://www.nature.com/articles/nmeth.3688>, <https://www.ncbi.nlm.nih.gov/pmc/articles/PMC3817102/>) and, in our opinion, highlight the need for validation of candidate proteoforms derived from Ribo-seq analysis by alternative means. Following initial filtering, a second, orthogonal selection based on biological information was made to retain the potentially most interesting pairs. From this list, a final selection was made based on criteria to prioritize candidates suited for Virotrap, which meant prioritizing proteoforms with a substantial difference at their N-termini, non-structural proteins and proteoforms with losses or gains of protein domains or motifs. This likely also introduced a bias in our results as these criteria might increase our chances of identifying differences between the interactomes of the studied proteoforms. As such, it will likely be error-prone to translate our results towards general conclusions about N-terminal proteoforms.”

The authors could comment or explain the discrepancy in the number of N-terminal proteoforms identified in their study compared to the literature estimates.

We suspect that the reviewer meant that we reported less N-terminal proteoforms (such as those originating from 5'UTR translation) than Ribo-seq based studies. However, the number of N-terminal proteoforms detected in our study corresponds to numbers detected by similar N-terminal proteomics approaches used to identify N-terminal proteoforms originating from different translation events (e.g. <https://pubmed.ncbi.nlm.nih.gov/24623590/>). On the other hand, studies also focusing on neo N-termini originating from processing – which, we like to repeat was not the aim of our current study – reported on more N-terminal proteoforms (e.g. <https://pubs.acs.org/doi/full/10.1021/acs.jproteome.5b00579>, <https://www.ncbi.nlm.nih.gov/pmc/articles/PMC3872078/>), yet less N-terminal proteoforms originating from differently spliced transcripts and alternative translation initiation.

As mentioned in our answer to the previous comment, we hypothesize that our stringent filtering strategy to some extent accounted for the discrepancy compared to studies that also considered processing events. Indeed, when re-examining our data from trypsin-digested samples searched in our custom database, we found that when considering all unique N-terminal peptides (thus, only considering Ace- or AcD4-starting peptides), 5,062 peptides were identified, of which 3,600 mapped to a position beyond the second position in the protein sequence. Of these, 468 were retained after filtering for translation products, implying that we filtered out 3,132 N-terminal peptides potentially originating from proteolytic processing, which is in line with data reported in the abovementioned studies.

When evaluating the distribution of identified accessions before any filtering step, distributions of N-terminal proteoforms similar to the composition of our custom database (see below or <https://pubmed.ncbi.nlm.nih.gov/35788065/> supplementary figure S3, left side pie charts) are found. However, after the first filtering step (accession sorting), this distribution disappears as

almost all peptides get linked to UniProt entries. This first filter is crucial as peptides often match multiple protein sequences, both well-annotated one as well as novel ones. As Mascot was used at the peptide level, database entries seem to have been listed pretty much at random following identification of such multi-protein matching peptides, this as Mascot deals with the protein inference problem at the protein level. To correct for this, we reordered all peptide-associated protein entries, prioritizing UniProt entries over UniProt isoform entries, and these over Ensembl entries (coming from the Ribo-Seq data) as the UniProt database is by far the most completely annotated and curated of these databases. This thus shows that the discrepancy is due to our stringent filtering approach, but also that our filtering approach is necessary to report true protein evidence of these alternative ORFs.

Additionally, also see the answer provided to the previous question, where we make a comparison with database composition and with previous riboproteogenomics studies.

Figure S3: Pie chart showing the distribution of all types of accession for the trypsin digested sample before (A) and after accession sorting (B), for the chymotrypsin digested sample before (C) and after accession sorting (D) and for the GluC digested sample before (E) and after (F) accession sorting.

The authors could be careful when blaming the low agreement between Y2H and protein capture techniques on Y2H, since Y2H is the vivo technique.

While Y2H is performed in living cells, there are several limitations inherent to Y2H that might hamper detection of prey proteins. These limitations arise from overexpression of human proteins/proteoforms in a yeast cell, the possibility that bait and prey proteins do not carry the same co- and post-translational profiles in yeast cells compared to human cells (and such profiles may steer PPIs), and the fact that bait-prey interactions are forced to occur in the yeast's nucleus. On the other hand, Virotrap avoids a large drawback found in many AP-MS studies as protein complexes are extracted in VLPs and herein conserved before lysis and preparations for LC-MS/MS take place.

To address this comment, we adjusted the below text (text in bold was added and text in strikethrough presentation was removed) in the discussion session (page 55).

*“Validation of Virotrap data was ~~attempted~~ **done** by Y2H on all 20 pairs of proteoforms. ~~However, as only a total of 39 PPIs were found by Y2H, we could not provide a lot extra evidence supporting our Virotrap findings.~~ **Y2H identified 39 PPIs providing extra evidence supporting our Virotrap data.** The lower number of PPIs found by Y2H could **partially** be explained by the technology's limitations **and the considered approach, as baits were only tested in one set-up (bait coupled to AD and tested for interaction with prey coupled to DB, and not in other set-ups such as bait coupled C-terminally to AD or bait coupled to DB).** **Therefore, we found that Y2H was ~~reported to~~ did not work for a the majority of baits, also likely, due to auto-activation or failure of nuclear localization of bait and/or prey, and typically reports strong PPIs [40, 71].** Nevertheless, we were able to validate the interaction of PDCD4 with the full-length EIF4A1 protein, reported by Virotrap.”*

Minor points

Minor and technical comments (including suggestions to the Authors, so that they could improve the presentation and accessibility of their manuscript):

The authors could use consistent language and abbreviation throughout the paper, especially for N-terminal / Nt and N-terminal proteoform / Nt-proteoform.

We thank the reviewer for pointing this out and we changed all use of Nt- back to N-terminal. Other abbreviations were also checked and adjusted to always use the abbreviation once it was introduced.

Abbreviations could be introduced in the Results section as well as in the Materials and Methods section.

Abbreviations were now also used in the Materials and methods section. From the author guidelines of Life Science Alliance, we noticed that the Material and Methods sections comes almost at the end. All terms were thus introduced prior to this section, if possible. Abbreviations first used in the Materials and Methods section were properly introduced and used consistently throughout the manuscript.

The resolution of figures is low, and the color schemes could be friendlier, especially in Figure 6.

We assume low figure quality is only due to the embedding in the PDF and figures will be uploaded separately in high resolution. The figures were revised by all co-authors and several lab members and no comments were made on this, therefore, we kept the color schemes. In fact, it was noticed that the color scheme allowed for easy interpretation and was used consistent over all figures.

Supplementary tables could be named the same as in the manuscript.

We always refer to Supplementary tables as Supplementary Table SX in the text and also use this format in the naming of the Excel files. We thus do not know how we should adjust the names as they already seem uniform. There might have been introduced a discrepancy in the merging of all files. We doublechecked the naming in the text and the names of the files when uploading the revised version and found no inconsistencies.

Abstract:

The abstract is well written and provides a good overview of the study aim and what was done. However, the authors could elaborate on how Nt-proteoforms originate from splice variants in the introduction section.

In the introduction, the following part was adjusted in response to this request (text in bold was added):

*“Studies in our lab revealed that 10-20% of protein N-termini in several human and mouse cells point to alternative translation initiation and/or alternative splicing [5]. Such N-terminal proteoforms thus stem from the same gene but differ at their N-terminus. **In eukaryotes, the canonical mechanism for translation to start involves a ribosome assembling at the 5’ end of a mature mRNA molecule, which then starts scanning for start codons towards the 3’ end. Alternative start codons can be used for translation by various mechanisms such as leaky scanning or internal ribosome entry sites. In addition, alternative splicing may give rise to transcripts that have different 5’ ends (e.g. due to skipping of the first exon) [ref.]**The majority of N-terminal proteoforms are truncated at the N-terminus relative to the canonical form however, up to 6% have extended N-terminal regions presumably caused by ribosomes starting translation from codons in the annotated 5’UTR. N-terminal proteoforms can also carry modified N-termini different from those of the canonical protein [2, 6, 7].”*

Introduction:

The topic is introduced well, and the reader understands the aim and scope of the study, as well as why the topic is of interest. However, some aspects to understand the workflow are not covered well. COFRADIC and Virotrap are mentioned and linked to literature, but they are key parts of the workflow and could be explained in a few sentences.

The introduction was adjusted to:

*“Here, we first applied N-terminal COFRADIC [45] on the cytosol of HEK293T cells to construct a comprehensive catalogue of N-terminal cytosolic proteoforms. **N-terminal COFRADIC in essence relies on two consecutive, identical chromatographic separations of peptides, interrupted by a chemical reaction with 2,4,6-trinitrobenzenesulfonic acid (TNBS) causing a hydrophobic shift of internal peptides which is exploited to capture N-terminal peptides during the second chromatographic separation [45].** ~~We then applied stringent filtering to~~*

~~select proteoforms pairs for interactome analysis by Virotrap (see Figure 1), a method to study protein-protein interactions that avoids cell lysis by exploiting the characteristics of the HIV-1 p55 GAG protein which leads to the production of virus-like particles (VLPs). We then applied stringent filtering to select proteoforms pairs for interactome analysis by Virotrap (see Figure 1). In short, in Virotrap, a bait protein is fused to the C-terminus of the HIV-1 GAG protein, leading to the recruitment of the GAG-bait fusion protein at the plasma membrane where GAG multimerization occurs, followed by subsequent budding of virus-like particles (VLPs) from the cells. As the bait is coupled to GAG, this allows for co-purification of bait-associated protein partners by trapping them into VLPs. Purification of the VLPs themselves relies on co-expressing FLAG-tagged and untagged VSV-G, presented as trimers on the surface of VLPs, allowing for efficient antibody-based purification of the VLPs. Of note, Virotrap was shown to be a sensitive PPI method, as VLPs encapsulate and preserve the protein complexes, allowing the detection of weak and transient protein-protein interactions [46].”~~

The first three paragraphs are also repetitive in language and need rewording.

~~We have made some adaptations through the text to avoid this repetitive wording. In the following parts, the strikethrough text was removed.~~

~~“Eukaryotic protein-coding genes give rise to several protein variants, or proteoforms, through various mechanisms including genetic alterations, alternative promotor usage during transcription and alternative splicing during mRNA maturation, use of alternative initiation codons and stop codon read-through during translation,, numerous co- and post-translational modifications [1-3] Crosstalk between these mechanisms greatly expands a proteome’s complexity [4]. Studies of our lab revealed that 10-20% of protein N-termini in several human and mouse cells point to alternative translation initiation and/or alternative splicing [5]. Such N-terminal proteoforms thus stem from the same gene but differ at their N-terminus. In eukaryotes, the canonical mechanism for translation to start involves a ribosome assembling at the 5’ end of a mature mRNA molecule, which then starts scanning for start codons towards the 3’ end. Alternative start codons can be used for translation by various mechanisms such as upon leaky scanning or the use internal ribosome entry sites [6, 7]. In addition, alternative splicing may give rise to transcripts that have different 5’ ends (e.g. due to skipping of the first exon) [8, 9]. The majority of N-terminal proteoforms are truncated at the N-terminus relative to the canonical form however, up to 6% have extended N-terminal regions presumably caused by ribosomes starting translation starting from codons in the annotated 5’UTR. N-terminal proteoforms can also carry modified N-termini different from those of the canonical protein [2, 10, 11].~~

~~N-terminal proteoforms are often overlooked and information on their biological function is often based on atomistic studies focusing on one gene [2, 9, 12-14]. N-terminal proteoforms They may have different functions as the N-terminus of a protein steers several protein features such as half-life and protein localization [15, 16]. Concerning the latter, many targeting signals reside at a protein’s N-terminus and, consequently, N-terminally truncated or extended proteoforms may lose or gain targeting signals, causing such proteoforms to reside at different subcellular localizations [12, 17-25]. Several N-terminal proteoforms with such altered subcellular localization are These might be iso-functional, but thus and thus active in different compartments [19, 20]. Our lab and the Kuster lab showed that pairs of N-terminal proteoforms originating from the same~~

~~gene can possess different stabilities in cells [9, 26, 27]. Of note, mounting evidence indicates that alternative translation initiation is regulated in response to a variety of stress stimuli and/or in a tissue and a cell developmental specific manner [3, 28, 29]. Van Damme et al. (2014) also showed that alternative translation initiation sites are generally conserved among eukaryotes, hinting to their possible biological impact [5]. In addition, several N-terminal proteoforms have already been linked to human diseases, illustrating their potential for therapeutic intervention, diagnosing and prognosing disease [2, 30].~~

~~Other studies showed that N-terminal proteoforms may have altered functionalities [14, 28, 29, 31-36]. An example is the regulator of G-protein signaling (RGS2) which was reported to give rise to four different N-terminal proteoforms starting at methionines 1, 5, 16 or 33. The proteoforms starting at positions 16 or 33 have an impaired inhibitory effect on type V adenylyl cyclase (ACV) compared to the full-length protein and it was suggested that these N-terminal RGS2 proteoforms are part of a novel negative feedback control pathway for adenylyl cyclase signaling [32]. Different studies illustrated that N-terminal proteoforms can interact with proteins other than the interaction partners of their canonical protein. For example, the fibroblast growth factor-2 (FGF-2) exists in multiple proteoforms: a low molecular weight N-terminal proteoform (18 kDa) generated upon alternative usage of a start codon, and at least two higher molecular weight proteoforms (21 and 23 kDa) generated upon translation starting from CUG codons located in the 5' UTR of the corresponding transcript. The 18 kDa and 23 kDa proteoforms have different localizations and different functionalities. Moreover, the 23 kDa FGF-2 proteoform co-immunoprecipitated with the survival of motor neuron protein (SMN), whereas the 18 kDa proteoform did not. The authors hence concluded that SMN specifically interacts with the 23-kDa FGF-2 proteoform by binding to its N-terminal extension [14].”~~

Additionally, the statement "Nt-proteoforms have long been overlooked and studies on their biological function are emerging just now" needs revision since the cited references are more than 20 years old.

~~One reference [11] is indeed more than 20 years old, yet we wanted to state that N-terminal proteoforms are not often considered for functional studies and that the information we now have about their biological roles is mainly based on individual studies of just one proteoform. We agree with the comment of the reviewer and adjusted the sentence to the following (see below), to better suit the information/ statement we intend to give.~~

~~The sentence was adjusted to: "N-terminal proteoforms are often overlooked and information on their biological function is often based on atomistic studies focusing on one gene. “~~

Materials and Methods:

The analysis code should be made available, and the generation of a cytosolic proteome map of HEK293T cells needs better explanation.

~~The analysis code was made available in the following Github repositories: <https://github.com/dfjkw/Nt-proteoform-R-workflow.git> and <https://github.com/dfjkw/Tissue-expression-Nt-proteoforms.git>.~~

~~The Materials and Methods section contains a section entitled “Generation of a cytosolic proteome map of HEK293T cells”, which describes all the experimental work and data analysis~~

to the detail so that it can be fully reproduced by other researchers. Hence, we are not sure what information is missing or should be added.

The authors could clarify how protein/peptide MS quantities were normalized across different tissues when evaluating tissue expression of Nt-proteoforms through re-analysis of public proteomics data.

Our method of quantitative data analysis was based on NSAF values (normalized spectral abundance factor) that account for protein size and variability between runs. SAF values were first calculated per protein based on total spectral counts corrected for protein length (since longer proteins produce more peptides). Next, to accurately account for variation between samples, individual SAF values were normalized to one by dividing by the sum of all SAFs per sample, resulting in the NSAF value. In analogy to microarray and TMT data analysis, NSAF values were thereby standardized comparable between samples and runs. Log transformation of NSAF values is then typically performed to achieve normal distribution of data suitable for downstream parametric statistical analysis methods (in our case, a moderated t-test using the limma package).

The sentences in bold were added to the appropriate materials and methods section:

*“Per sample and replicate, we obtained a unique peptide count, spectral count and NSAF (normalized spectral abundance factor) quantification. **Briefly, SAF values were first calculated per protein based on total spectral counts corrected for protein length (since longer proteins produce more peptides). Next, to accurately account for variation between samples, individual SAF values were divided by the sum of all SAFs per sample, resulting in the NSAF value. Log transformation of NSAF values was performed to achieve normal distribution of data suitable for downstream statistical analysis. Differential expression analysis across all tissues was performed using limma (3.50.3) based on log2NSAF values...**”*

The authors could also introduce abbreviations in the Materials and Methods section, as some were only introduced in the Results section.

See above. We double-checked the text and adjusted it where needed.

For Virotrap studies, the authors could introduce the abbreviations PR and FL at their first use, clarify what was measured, and explain when imputation was applied and why. The authors could also clarify if peptide quantities were normalized between baits and control and why missing potential interaction partners of all baits were imputed.

Introduction of FL and PR term in part of Materials and Methods: Generation of Virotrap clones in the following sentences (text in bold was added):

*“Gag-bait fusion constructs were generated as described [46]. The coding sequences for the full-length protein (**referred to as FL**) were either ordered from IDT (gBlocks gene fragments, as was the case for the following constructs: CAPRIN1, SPAST, PRUNE, SORBS3, FNTA, CAST, RARS, UBXN6, PAIP1, PXN and NTR protein) or generated cDNA was used as template for PCR amplification using AccuPrime™ pfx DNA polymerase (Invitrogen) and ORF-specific primers. cDNA was generated by isolating RNA from 5x10⁶ HEK293T cells with the Nucleospin RNA isolation Mini kit (Macherey-Nagel) according to the manufacturer’s instructions. 500 ng of isolated RNA was then used as input for generating cDNA and cDNA synthesis was performed using the PrimeScript RT kit (Takara Bio) according to the manufacturer’s instructions. The generated PCR products for the full-length proteins were transferred into the pMET7-GAG-sp1-RAS plasmid by*

classic cloning with restriction enzymes (EcoRI and XbaI) or In-Fusion seamless cloning (Takara Bio) when the genes contained internal restriction sites for EcoRI and XbaI (which was the case for CSDE1, NTR and UBAC1).

The N-terminal proteoforms (**referred to as PR**) were amplified from the corresponding generated pMET7-GAG-sp1-FL plasmid of each gene using the AccuPrime™ pfx DNA polymerase (Invitrogen) with proteoform-specific primers.”

We are not sure where we need to specify “what was measured”. We assume that in the sections describing the Virotrap studies it was insufficiently stated that the VLPs (containing bait and its associated protein interaction partners) are enriched from the cellular supernatant and subsequently lysed to release the entrapped proteins. Proteins are then digested by trypsin and the resulting peptide mixtures are analyzed by LC-MS/MS. Each set consists of triplicate control samples and two or three pairs of baits (a pair of one bait is thus the combination of triplicate samples of bait A FL and triplicate samples of bait A PR). A small adaptation was made in the Materials and Method section to make this more clear:

“The cellular supernatant (**containing the VLPs that engulf the bait and its associated interaction partners**) was harvested after 46 h and centrifuged for 3 min at 1,250 x g to remove debris.”

“Beads were added to the samples and the VLPs were allowed to bind for 2 h at room temperature by end-over-end rotation. Bead-particle complexes were washed once with 200 µl washing buffer (20 mM Tris-HCl pH 7.5 and 150 mM NaCl) and subsequently eluted with FLAG peptide (30 min at 37 °C; 200 µg/ml in washing buffer). **VLPs were and** lysed by addition of Amphipol A8–35 (Anatrace) [60] to a final concentration of 1 mg/ml. After 10 min, the lysates were acidified (pH <3) by adding 2.5% formic acid (FA). Samples were centrifuged for 10 min at >20,000 x g to pellet the protein/Amphipol A8–35 complexes. The supernatant was removed and the pellet (**containing the proteins**) was resuspended in 20 µl 50 mM fresh triethylammonium bicarbonate (TEAB). Proteins were heated at 95 °C for 5 min, cooled on ice to room temperature for 5 min and digested **into peptides** overnight at 37 °C with 0.5 µg of sequencing-grade trypsin (Promega). Peptide mixtures were acidified to pH 3 with 1.5 µl 5% FA. Samples were centrifuged for 10 min at 20,000 x g and 7.5 µl **of the sample** was injected for LC-MS/MS analysis on an Ultimate 3000 RSLCnano system in-line connected to a Q Exactive HF Biopharma mass spectrometer (Thermo Scientific). Details on the LC-MS/MS settings and data-analysis can be found in Supplementary Materials and Methods.”

The supplementary Materials and Methods section already indicated when imputation was applied:

“Proteins only identified by site and reverse database hits were removed as well as potential contaminants. Replicate samples were grouped and proteins with less than three valid values in at least one group were removed. Missing values were imputed using imputeLCMD (R package implemented in Perseus) with a truncated distribution with parameters estimated using quantile regression (QRILC).”

Imputation was thus performed after filtering on valid values.

In response to why we perform imputation we would like to state the following:

First of all, we want to point out that there is no standardized approach of analyzing Virotrap or interactomics data and, that in our case, we were confronted with more comparisons than usual as we compared FL versus control, PR versus control and FL versus PR. A major drawback of DDA and LC-MS/MS analysis in general is the significant number of missing values, which hinder proper statistical data analysis. Imputation of missing values is thus essential for the identification of “regulated proteins” (here, preys) in the different samples. However, we are aware that we combined different baits per analyzed set, and thus specific preys of one bait will or should be absent in samples in which other baits were studied. Imputation of missing values of such specific proteins in all samples could cause preys not being significant anymore. However, as we expect strong interaction partners to be abundantly present in VLPs and missing values are imputed with lower LFQ intensities, we should still be able to report such proteins as significant. If the intensities of a potential prey are similar to the imputed value, such a prey is less likely to be a strong interaction partner. Following this reasoning, we are aware that we will have missed potential interaction partners due to imputation. However, to control the effect of imputation, we checked reported preys using intensity profile plots before imputation. Additionally, one could also run a script to identify all clear cases where a protein is found in all bait samples and absent in all control samples, or present in all PR samples and absent in all FL samples or vice versa. On the other hand, as imputation was done for all baits in one set which also allowed us to better identify the background proteins present the VLPs, further allowing us to discriminate between proteins with lower levels in eDHFR control samples from preys of a given bait.

Concerning the comment on normalization, besides the normalization function embedded in the MaxLFQ algorithm, no additional normalization was performed neither on the peptide or on the protein level.

Very similar comments were made on the results section and we also would like to refer to our answers given below.

Results:

In the Interaction profiling of Nt-proteoforms and their canonical counterparts section, the authors could address MaxQuant's normalization strategies and how similar VLP contents were within VLPs. They could also clarify if VLPs of different baits can be analyzed together regarding normalization and imputation.

In MaxQuant, the normalization factors are optimized to achieve the least overall proteome variations. Thus, this requires a population of identified proteins that changes minimally between the different samples. Our experiments show, that besides the interaction partners and baits, many “background proteins” are consistently identified in the VLPs. In fact, 458/1997 (23%) of proteins are identified in 80% of Virotrap samples, providing a basis for MaxLFQ normalization. The normalization process has a moderate effect on overall protein intensities. Median log2 intensity per sample is affected by 1.3%, on average (Mean absolute percentage difference). To additionally verify the presence of a stable background, we checked profile plots of common VLP proteins, namely several plasma membrane proteins (before imputation) such as: ATP1A1, ATP1B3 and several SLC proteins (SLC16A1, SLC1A5, SLC29A1, SLC3A2, SLC7A5) as well as several proteasome proteins such as PSMC1 until PSMC6, PSMD1 until 14. We detected stable protein intensities over all samples and almost no missing values. Indicating that these VLP proteins provide a good background for MaxLFQ normalization, also between baits.

We therefore decided to analyze different baits together, provided they were run on the mass spectrometer at the same time (in one batch or “set”). However, to avoid the need for batch correction, we are skeptical of analyzing different sets together. Interactomics studies are in general delicate and introducing extra variance is unnecessary.

The following sentence (indicated in bold) was added to the text:

*“For LC-MS/MS analysis, we divided the baits in manageable sets including control samples and maximally three pairs of FL and PR. Here, baits with similar expression levels were combined and triplicate experiments for all baits were performed. In total, baits were divided into seven sets (see Materials and Methods section). To obtain specific interaction partners of the FL or PR from the lists of identified proteins, their interactomes were compared with that of the control baits. To evaluate possible functional differences between FL and PR, the identified proteins using the FL baits were directly compared with those from the PR baits (Figure 4.B). **Our experiments show, that besides the interaction partners and baits, many “background proteins” are consistently identified in the VLPs. In fact, for example in the first set 458/1997 (23%) of proteins are identified in 80% of Virotrap samples, providing a basis for MaxLFQ normalization.**”*

For the Proteoform-specific interaction partners section, the authors could explain the reason for imputation and whether it would be necessary for a more suitable analysis strategy.

There are several ways of analyzing affinity purification MS data and according to our knowledge, there is no golden standard approach. In our study, the imputed values were systematically small, adjusted to the low-end of protein intensities observed in each sample. Imputation was only performed for proteins identified in $\frac{3}{4}$ replicates of one condition. This allowed for statistical analysis of proteins unique to a given bait. An alternative strategy, without imputation, would be to obtain a statistical test results when possible and additionally consider unique proteins as interactors, which is also not ideal.

In the Studying selected differences between proteoform interactomes by AP-MS section, the authors could clarify why some proteins did not show up in the table of candidate interactors for AP-MS but were present in the Volcano plot.

There are three proteins: IFIT5, PNPT1 and APOL2 that were reported as significant proteins between the FL and PR but that are not reported as candidate interaction partners. Upon checking the profile plots of these proteins before imputation (see profile plot below, APOL2 in red, PNPT1 in blue and IFIT5 in black), one notices that all these proteins are most intense in MAVS FL samples and especially APOL2 and IFIT5 seem specific for MAVS FL as they are not identified in many other samples. However, all these proteins are identified at quite low intensities and thus do not seem strong interaction partners but their profile suggest that they could be (presence in MAVS FL samples and absent in most other samples or more abundantly present specifically in MAVS FL samples). They indeed also seem to differ between the FL and PR and we are thus not surprised that these proteins are suggested as interaction partners in the volcano plot comparison between FL-PR.

We assume that these proteins might be not reported as interaction partners due to imputation. We therefore check their profile after imputation (see below). We see especially for APOL2 and IFIT5 a large difference in the imputed values causing a highly variable profile. This could cause the protein not to be considered as significant.

This shows that our approach is not fault-proof, but as mentioned above, we still believe that our approach remains valid and we likely reported the most confident interactors. Currently no general or standard approach exist for the analysis of PPI datasets. One approach that could have helped to report APOL2 and IFIT5 as potential interaction partners would be to include black-white cases (presence/absence).

They could also evaluate the comparison of identified interaction networks to known interaction networks as mentioned in the text.

We already covered the inclusion of known interactions as we discussed the overlap with interaction partners reported in known databases at several points in the Results section. For example, this was always reported in the tables (e.g. Table 4) and is briefly discussed in the text. For example: “We compared our list of potential interaction partners with known interactors listed in BioGRID [53], STRING [41] and IntAct [62], and with candidate interactors identified by Virotrap. Six out of 14 candidate interaction partners were reported in at least one of the consulted databases, while just two candidates were also reported by Virotrap. The rather high overlap with known interaction partners points to the quality of the AP-MS data, while the overlap with the Virotrap results is a bit lower.”

In response to this comment, the following sentences (indicated in bold) were added in the last paragraph:

*“Based on our Virotrap and AP-MS interactomics data for both the canonical protein and N-terminal proteoform of MAVS and cross-checked with known interactors listed in BioGRID [53], STRING [41] and IntAct [62], we generated a protein-protein interaction network of MAVS (Figure 8), showing all identified candidate interaction partners. Each edge represent an identified interaction between the bait (either MAVS FL or PR, red nodes) and prey protein. In total, we identified 65 proteins and 115 interactions. The majority of the interaction partners (38) are shared between the canonical protein and the N-terminal proteoform (clustered in the middle). However, for both MAVS FL and PR, we also report a set of unique interaction partners (clustered on the left and right side). For the canonical protein, we found 16 preys that to solely interact with the canonical protein, while for the proteoform we report nine unique interaction partners. **From all identified potential interaction partners, nine are known interaction partners listed in at least one of the three consulted PPI databases. This shows that both Virotrap and AP-MS identified known interaction partners and, besides these, both approaches also reported several novel potential interaction partners, many of which are only identified with one of the two analysis methods. This demonstrates the uniqueness of each PPI method and the difficulties often encountered when validating specific interactions. Additionally, all unique or specific interaction partners reported for MAVS PR are not reported in either BioGRID, STRING or IntAct. This could hint to an alternative function of the proteoform which is unrelated to the function of the FL protein.**”*

Lastly, the last paragraphs of the Interaction profiling of Nt-proteoforms and their canonical counterparts section are not well-formulated.

We think that the reviewer is pointing to the paragraphs below, which we attempted to make more understandable.

*“Some **selected interesting** findings on proteoform-specific interactors are discussed in the following section.*

*For MAVS, ~~a~~ the pairwise test between **the interactomes of** FL and PR resulted in 10 significant proteins (Figure 7.A), ~~being:~~ PTK7, PLK1, HNRNPUL2, CEP55, CD2AP, ARRDC1, DAG1, EIF3K, TRAF2 and NUP205. ~~Of these, only~~ TRAF2 and PLK1 have been reported above as candidate interaction partners for MAVS FL and PR respectively. ~~, and are thus withheld as proteins that possibly interact differently with the MAVS proteoforms. By this stringent filtering, we thus remove several proteins that, although they seem to interact differently with MAVS FL or PR, they are unlikely to be interaction partners. In fact,~~ **This is also** obvious by visualizing ~~from~~ their intensities profiles in the different samples before imputation (Figure 7.B), which show that TRAF2 and PLK1 interact with MAVS with a significant difference in intensity ~~between~~ **in** the FL and PR interactomes. For comparison, we also show the profile of two proteins that were reported as significantly different between the FL and PR interactomes, but were removed as **they** were not listed as candidate interactors. A first example is DAG1, which was not identified in any of the MAVS interactomes, ~~but while the found~~ difference in the interactomes of the MAVS proteoforms is ~~only~~ due to differences in ~~the~~ imputed intensity values. In fact, this ~~example~~ highlights an imputation-based shortcoming of ~~the~~ data analysis software however, imputation is necessary for statistical analysis. A second example is ARRDC1, which is identified in almost all samples with an apparent lower intensity in the MAVS samples, making it thus unlikely that ARRDC1 is an*

interaction partner of MAVS. To conclude, it seems that MAVS PR ~~has lost the interaction~~ **does not interact** with TRAF2, which could be due to the fact that ~~in MAVS PR, the domain required for this interaction with TRAF2 becomes outer~~ **is exposed at the N-terminal part of MAVS PR** (Figure 5.A), ~~which affects the interaction with TRAF2~~. On the other hand, MAVS PR seems to interact ~~better~~ **stronger** with PLK1, suggesting that MAVS PR ~~proteoforms can both gain and lose~~ **as well as gain** interactors.

For the polyadenylate-binding protein-interacting protein 1 (PAIP1), an N-terminal proteoform starting at position 113 **which is a** known UniProt isoform, was detected. This PAIP1 N-terminal proteoform specifically interacts with GIGYF2, ZNF598 and EIF4E2, which together form the 4EHP-GYF2 complex. GIGYF2 and ZNF598 were identified from the comparison of the FL and the PR interactomes, while ZNF598 and EIF4E2 were listed as candidate interactors of PAIP1 PR (Figure 7.C). The engagement of PAIP1 PR with the 4EHP-GYF2 protein complex could point to a different functionality of the PAIP1N-terminal proteoform versus full-length PAIP1.

~~Opposite to PAIP1 PR, we report that~~ **on the other hand** the N-terminal proteoform (missing amino acid 1-211) of the eukaryotic initiation factor 4A-I (EIF4A1) loses several known interactions (see Figure 7.D). In the pairwise comparison between FL and PR, we found that the interaction with eight candidate eukaryotic initiation factors (EIF4A3, EIF4E, EIF4G3, EIF4B, EIF4G2, EIF4A2, EIF4H and EIF4G1), ~~seems to be~~ **is** specific for the FL, while on the side of the EIF4A1 proteoform, we identified ~~amongst the significant candidate interaction partners~~ several proteins involved in proteasome-mediated protein degradation **as possible interactors**.

5. Y2H screens to validate the interaction profile of N-terminal proteoforms

To support our previous findings on the interactome of N-terminal proteoforms, we performed a Yeast two-hybrid (Y2H) screen ~~similar~~ as reported in [8, 63]. ~~Y2H~~ **These** screens were performed ~~in which~~ **such that** all baits (both canonical protein and N-terminal proteoform) **were** fused to the Gal4 DNA binding domain (DB), ~~were~~ **and** tested against proteins encoded by the hORFeome v9.1 collection containing 17,408 ORF clones fused to the Gal4 activation domain (AD). Following first-pass screening, each bait was pairwise tested for interaction with all the candidate partners identified for any proteoform of that gene. Pairs showing a positive result were subjected to a pairwise retest and PCR products amplified from the final positive pairs were sequenced to confirm the identity of clones encoding each interacting protein. This resulted in the identification of 39 high confident binary protein-protein interactions PPIs (~~listed in~~ Supplementary Table S9). Note that not for all baits PPIs are reported, which is due to the auto-activation of the reporter gene for some baits, while for other baits no interactions were found or did not result in a positive pair after pairwise tests.

Out of the 39 high confidence binary PPI's reported with Y2H, 11 (28.2 %) are also reported as candidate interaction partners by Virotrap (~~indicated in~~ Supplementary Table S9). As in general the overlap between different PPI methods is not ~~so~~ high [68-70], this relative high overlap shows the quality of both datasets. As an example, for EIF4A1, the specific interaction of the canonical protein (and not the PR) with PDCD4 (a well-known interaction partner), is supported by both Virotrap and Y2H.

6. Studying selected differences between proteoform interactomes by AP-MS

We selected three baits, MAVS, EIF4A1 and PAIP1, for further validation by affinity purification mass spectrometry (AP-MS). Both FL- and PR-bait-FLAG fusion constructs were generated in which FLAG is fused to the C-terminus of the bait to avoid steric hindrance of the tag on the bait's N-terminus. Four biological repeats of pull-down experiments using FLAG-tagged baits and an eDHFR-FLAG control were performed. Quantitative mass spectrometry was used to quantify the interaction partners of all proteoforms. In total, 1,903 proteins were identified over all experiments (Supplementary Table S10). Pairwise contrasts between control-bait samples and between proteoform samples were selected at a Benjamini–Hochberg adjusted p -value (FDR) ≤ 0.05 (Supplementary Table S10, second tab). Such tests between eDHFR control samples and baits resulted in 14 candidate interaction partners for MAVS FL, while for MAVS PR, only the bait was found as being significant (see Table 4 and Supplementary Table S10, second tab).

We compared our list of potential interaction partners with known interactors listed in BioGRID [53], STRING [41] and IntAct [62], and with candidate interactors identified by Virotrap. Six out of 14 candidate interaction partners were reported in at least one of the consulted databases, while just two candidates were also reported by Virotrap. The rather high overlap with known interaction partners points to the quality of the AP-MS data, ~~while the overlap with the Virotrap results is a bit lower.~~

When applying the same selection criteria as used in the pairwise tests between Virotrap data for FL and PR, the pairwise comparison between the MAVS FL and PR interactomes reveals several candidate interaction partners that are enriched in the MAVS FL interactome (Figure 7.E and Supplementary Table S10, tab 4). In fact, our AP-MS data support our Virotrap findings that TRAF2 is enriched in MAVS FL interactomes. ~~and thus that this interaction is affected for the N-terminal proteoform. Virotrap also reported the specific interaction of MAVS PR with PLK1 however, PLK1 was not identified in our AP-MS study.~~

For PAIP1, we identified several candidate interactors of both FL and PR, nine and 12 respectively, of which seven are known interaction partners and two were also reported as interaction partners by Virotrap (Supplementary Table S10). However, none of the members of the 4EHP-GYF2 complex were found as candidate interactions partners ~~or as different between the FL and PR interactomes.~~ We could thus not support these specific Virotrap findings by AP-MS. We hypothesize that this could be due to the differences in the PPI techniques as Virotrap allows the detection of weaker and transient interactions due to the avidity effect of multiple bait copies lining the inside of the VLP. ~~The A pairwise comparison between PAIP1 FL and PR resulted in one significant protein, UBE2T, which is reported to be enriched in proteoform samples. However, this protein was not listed as a candidate interaction partner before.~~

For EIF4A1 FL, only two candidate interactors were found; EIF4A2 (known interaction partner, also reported by Virotrap) and IFNA2. For the corresponding PR, no candidate interactors could be identified. These two proteins were also reported as significant in the comparison between the FL and PR, with the proteins found to be specific for the FL **protein**. Our AP-MS data thus supports that the interaction of EIF4A1 with EIF4A2 is lost for the N-terminal proteoform, ~~as which was also reported found~~ by Virotrap.

Based on our Virotrap and AP-MS interactomics data for both the canonical protein and N-terminal proteoform of MAVS and cross-checked with known interactors listed in BioGRID [53], STRING [41] and IntAct [62], we generated a PPI network of MAVS (Figure 8), showing all identified candidate interaction partners. Each edge represent an identified interaction between

*the bait (either MAVS FL or PR, red nodes) and prey protein. In total, we identified 65 proteins and 115 interactions. The majority of the interaction partners (38) are shared between the canonical protein and the N-terminal proteoform (clustered in the middle). However, for both MAVS FL and PR, we also report a set of unique interaction partners (clustered on the left and right side). For the canonical protein, we found 16 preys that ~~to~~solely interact with the canonical protein, while for the proteoform we report nine unique interaction partners. From all identified potential interaction partners, nine are known interaction partners listed in at least one of the three consulted PPI databases. This shows that both Virotrap and AP-MS identified known interaction partners and, besides ~~these~~, **that** both approaches also reported several novel potential interaction partners, many of which are only identified with one of the two ~~analysis~~ methods. This **again** demonstrates the uniqueness of each PPI method and the difficulties often encountered when validating specific interactions. Additionally, all unique or specific interaction partners reported for MAVS PR are not reported in either BioGRID, STRING or IntAct. This could hint to an alternative function of the proteoform which is unrelated to the function of the FL protein.”*

Overall, the authors have done a lot of work and presented a valuable addition to the literature. While there are some issues in the details, they are solvable, and I recommend the publication of this work in LSA.

We thank the reviewer for her/his support of our manuscript.

May 25, 2023

RE: Life Science Alliance Manuscript #LSA-2023-01972-TR

Prof. Kris Gevaert
VIB-UGent Center for Medical Biotechnology
Center for Medical Biotechnology
Technologiepark-Zwijnaarde 75
9052, Ghent
Ghent 9052
Belgium

Dear Dr. Gevaert,

Thank you for submitting your revised manuscript entitled "N-terminal proteoforms may engage in different protein complexes". We would be happy to publish your paper in Life Science Alliance pending final revisions necessary to meet our formatting guidelines.

- please delete the contents section following the title page
- please consult our manuscript preparation guidelines <https://www.life-science-alliance.org/manuscript-prep> and make sure your manuscript sections are in the correct order
- please upload a clean manuscript without any track changes or highlighted text
- please add a conflict of interest statement to your main manuscript text
- please add a callout for Figure S4 to your main manuscript text
- in the Materials and Methods section, you write "See Supplementary Materials and Methods for additional information." I don't see this additional file, and in any case this should be incorporated into the main Materials and Methods section. We don't have a size limit on this section. If any unique References are mentioned there, please be sure to incorporate those into the main Reference list as well.

A. FINAL FILES:

B. MANUSCRIPT ORGANIZATION AND FORMATTING:

Thank you for your attention to these final processing requirements. Please revise and format the manuscript and upload materials within 5 days.

Sincerely,

May 30, 2023

RE: Life Science Alliance Manuscript #LSA-2023-01972-TRR

Prof. Kris Gevaert
VIB-UGent Center for Medical Biotechnology
Center for Medical Biotechnology
Technologiepark-Zwijnaarde 75
9052, Ghent
Ghent 9052
Belgium

Dear Dr. Gevaert,

Thank you for submitting your Research Article entitled "N-terminal proteoforms may engage in different protein complexes". It is a pleasure to let you know that your manuscript is now accepted for publication in Life Science Alliance. Congratulations on this interesting work.

DISTRIBUTION OF MATERIALS:

Again, congratulations on a very nice paper. I hope you found the review process to be constructive and are pleased with how the manuscript was handled editorially. We look forward to future exciting submissions from your lab.

Sincerely,
